# Cooperation of the ER-shaping proteins atlastin, lunapark, and reticulons to generate a tubular membrane network

Songyu Wang[1,2†], Hanna Tukachinsky[1,2†], Fabian B Romano[1,2†], Tom A Rapoport[1,2*]

[1]Howard Hughes Medical Institute, Harvard Medical School, Boston, United States; [2]Department of Cell Biology, Harvard Medical School, Boston, United States

**Abstract** In higher eukaryotes, the endoplasmic reticulum (ER) contains a network of membrane tubules, which transitions into sheets during mitosis. Network formation involves curvature-stabilizing proteins, including the reticulons (Rtns), as well as the membrane-fusing GTPase atlastin (ATL) and the lunapark protein (Lnp). Here, we have analyzed how these proteins cooperate. ATL is needed to not only form, but also maintain, the ER network. Maintenance requires a balance between ATL and Rtn, as too little ATL activity or too high Rtn4a concentrations cause ER fragmentation. Lnp only affects the abundance of three-way junctions and tubules. We suggest a model in which ATL-mediated fusion counteracts the instability of free tubule ends. ATL tethers and fuses tubules stabilized by the Rtns, and transiently sits in newly formed three-way junctions. Lnp subsequently moves into the junctional sheets and forms oligomers. Lnp is inactivated by mitotic phosphorylation, which contributes to the tubule-to-sheet conversion of the ER.

*For correspondence: tom_rapoport@hms.harvard.edu

†These authors contributed equally to this work

Competing interests: The authors declare that no competing interests exist.

## Introduction

The mechanisms by which organelles are shaped and remodeled are largely unknown. The endoplasmic reticulum (ER) is a particularly intriguing organelle, as it consists of morphologically distinct domains that change during differentiation and cell cycle. In interphase, the ER consists of the nuclear envelope and a connected peripheral network of tubules and interspersed sheets (*Shibata et al., 2009*; *Chen et al., 2013*; *English and Voeltz, 2013a*; *Goyal and Blackstone, 2013*). The network is dynamic, with tubules continuously forming, retracting, and sliding along one another. During mitosis in metazoans, the nuclear envelope disassembles and peripheral ER tubules are transformed into sheets (*Lu et al., 2009*; *Wang et al., 2013*). How the network is generated and maintained, and how its morphology changes during the cell cycle, is poorly understood.

Previous work has suggested that the tubules themselves are shaped by two evolutionarily conserved protein families, the reticulons (Rtns) and DP1/Yop1p (*Voeltz et al., 2006*). These are abundant membrane proteins that are both necessary and sufficient to generate tubules. Members of these families are found in all eukaryotic cells. The Rtns and DP1/Yop1p seem to stabilize the high membrane curvature seen in cross-sections of tubules and sheet edges (*Hu et al., 2008*; *Shibata et al., 2009*). How these proteins generate and stabilize membrane curvature is uncertain, but they all contain pairs of closely spaced trans-membrane segments and have an amphipathic helix that is required to generate tubules with reconstituted proteoliposomes (*Brady et al., 2015*). It has been proposed that the Rtns and DP1/Yop1p form wedges in the lipid bilayer and arc-shaped oligomers around the tubules (*Hu et al., 2008*; *Shibata et al., 2009*).

Connecting tubules into a network requires membrane fusion, which is mediated by membrane-anchored GTPases, the atlastins (ATLs) in metazoans and Sey1p and related proteins in yeast and

**eLife digest** The endoplasmic reticulum is a compartment within the cells of plants, animals and other eukaryotes. This compartment plays a number of roles within cells, for example, serving as the site where many proteins and fat molecules are built. Most often the endoplasmic reticulum exists as a network of thin tubules. However, this shape changes during the lifetime of a single cell, and the endoplasmic reticulum converts into flattened structures known as sheets when the cell divides.

Three classes of proteins are known to affect the shape of the endoplasmic reticulum. Proteins called reticulons (called Rtns for short) stabilize the highly curved membranes that make up the thin tubules, while proteins called atlastins (ATLs) fuse these tubules together to form the interconnected network. However, the exact role of the third protein – called lunapark (Lnp) – is unknown. Moreover, it is not clear how these three proteins work together to coordinate their individual activity to shape the endoplasmic reticulum.

Now, Wang, Tukachinsky, Romano et al. have used mammalian cells grown in the laboratory and extracts from the eggs of the frog *Xenopus laevis* to study these three proteins in more details. Unexpectedly, the experiments showed that ATL's activity was not only required to form a tubular network but also to maintain it. When ATL was inactivated, the network disassembled into small spheres called vesicles. Increasing the amount of Rtn within the endoplasmic reticulum also caused it to disassemble, but increasing the amount of ATL could reverse this fragmentation. Thus, maintaining the tubular network requires a balance between the activities of the ATL and Rtn proteins, with ATL appearing to tether and fuse tubules that are stabilized by the Rtns.

Wang et al. also found that the tubular network of the endoplasmic reticulum can form without Lnp, but fewer tubules and junctions are formed. These findings suggest that Lnp might act to stabilize the junctions between tubules.

Further experiments showed that Lnp is modified by the addition of phosphate groups before the cell begins to divide. Wang et al. propose that this modification switches Lnp off and helps the endoplasmic reticulum to convert into sheets. Further work is now needed to investigate exactly how Rtn, ATL, and Lnp shape the endoplasmic reticulum. These future experiments will likely have to use simpler systems, in which the purified proteins are incorporated into artificial membranes.

plants (*Hu et al., 2009*; *Orso et al., 2009*). These proteins contain a cytoplasmic GTPase domain, followed by a helical bundle, two closely spaced trans-membrane segments, and a cytoplasmic tail (*Bian et al., 2011*; *Byrnes and Sondermann, 2011*). Mammals have three isoforms of ATL, with ATL-1 being prominently expressed in neuronal cells. Mutations in ATL-1 can cause hereditary spastic paraplegia, a neurodegenerative disease that is characterized by the shortening of the axons in corticospinal motor neurons (*Salinas et al., 2008*). This leads to progressive spasticity and weakness of the lower limbs.

A role for ATL in membrane fusion is supported by the fact that proteoliposomes containing purified *Drosophila* ATL undergo GTP-dependent fusion in vitro (*Bian et al., 2011*; *Orso et al., 2009*). Furthermore, the fusion of ER vesicles in *Xenopus laevis* egg extracts is prevented by the addition of ATL antibodies or a cytosolic fragment of ATL (*Hu et al., 2009*; *Wang et al., 2013*). Finally, ATL-depleted *Drosophila* larvae have fragmented ER, and the depletion of ATL or expression of dominant-negative ATL mutants in tissue culture cells leads to long, unbranched tubules (*Hu et al., 2009*; *Orso et al., 2009*). Crystal structures and biochemical experiments have led to a model in which ATL molecules sitting in different membranes dimerize through their GTPase domains (trans-interaction), and undergo a conformational change during the GTPase cycle, thereby pulling the two membranes together and fusing them. Interestingly, ATL molecules sitting in the same membrane can also form GTPase-dependent dimers (cis-interaction) (*Liu et al., 2015*).

The fusion of membrane tubules generates three-way junctions, which are small, triangular sheets with negatively curved edge lines (*Shemesh et al., 2014*). How junctions are maintained is unknown, but it has been suggested that they are stabilized by the lunapark protein (Lnp), a conserved membrane protein (*Chen et al., 2012*; *Shemesh et al., 2014*; *Chen et al., 2015*). Lnp contains two closely spaced transmembrane segments and a $Zn^{2+}$-finger domain (*Chen et al., 2012*). Lnp localizes

preferentially to three-way junctions, but there is disagreement as to whether it stabilizes newly formed three-way junctions (in mammalian cells), or functions in ring closure, i.e. the disappearance of junctions (in yeast) (*Chen et al., 2012*; *Shemesh et al., 2014*; *Chen et al., 2015*). Furthermore, Sey1p and some isoforms of ATL also localize to three-way junctions and could therefore play a role in their stabilization (*Hu et al., 2009*; *English and Voeltz, 2013b*; *Yan et al., 2015*). Thus, the exact role of Lnp is unknown.

Although the available evidence suggests that Rtns/DP1, ATLs, and Lnp are major proteins responsible for determining ER morphology, it is unknown how they cooperate. For example, it is unclear whether one of these components acts upstream of another, whether they are all equally important to generate an ER network, and whether they have distinct roles in tubule or junction formation. It is also unknown whether any of these proteins are involved in the characteristic tubule-to-sheet conversion during mitosis.

Mammalian tissue culture cells offer the opportunity to study ER morphology by removing or overexpressing ER-shaping membrane proteins. So far, however, essentially all experiments employed fixed tissue culture cells that transiently express proteins or RNAi constructs, which results in poor viability, heterogeneity among cells, and fixation artifacts. Consequently, ER morphology changes are difficult to interpret. ER morphology can also be studied with extracts from *Xenopus laevis* eggs. Both crude extracts and isolated membranes can be used to form a tubular network in vitro (*Dreier and Rapoport, 2000*; *Wang et al., 2013*). With interphase extracts, the network consists exclusively of tubules connected by three-way junctions, whereas a tubule-to-sheet conversion and a loss of three-way junctions are observed with mitotic (meiotic) extracts (*Wang et al., 2013*). Because all known components involved in generating ER morphology are integral membrane proteins, their function can be tested with dominant-negative constructs, but not by depletion experiments.

Here, we have used live-cell imaging of stable mammalian tissue culture cells and *Xenopus* egg extracts to address how the known ER shaping proteins cooperate to generate and maintain a tubular ER network. We demonstrate that ATL function is not only required to form an ER network but, surprisingly, is also needed to maintain it. The integrity of tubules is maintained by a balance of ATL and Rtn function. Lnp is not essential for network formation, but its inactivation or absence causes the loss of three-way junctions and tubules in favor of sheets. Our results suggest mechanisms by which a tubular ER network is formed and maintained, as well as changed during mitosis.

## Results

### ATL is required to maintain tubules and junctions in mammalian cells

Previous experiments have shown that the transient expression of a dominant-negative ATL mutant in mammalian cells converts the ER network into long, unbranched tubules (*Hu et al., 2009*; *Rismanchi et al., 2008*). However, these experiments were performed with fixed overexpressing cells, in which it is difficult to exclude artifacts. We therefore revisited the effect of ATL and its mutant forms, using stable cell lines and live-cell imaging.

We first used CRISPR technology to generate U2OS cells that express at endogenous levels GFP-calreticulin, an established luminal ER marker. Live-cell imaging by spinning-disk confocal microscopy revealed both a tubular network and interdispersed sheet-like structures (*Figure 1A*; first column; *Figure 1—figure supplement 1A*). The GFP-calreticulin expressing cells were then infected with lentivirus constructs to generate stable cell lines that also express fluorescently-tagged ATL. The expression of wild type ATL isoforms had only a moderate effect on ER morphology (*Figure 1A*; second and third columns; *Figure 1—figure supplement1B,C*). Fluorescently tagged wild type ATL-1 localized throughout tubules (*Figure 1A*; second column), whereas wild type ATL-3 and ATL-2 localized in punctae at three-way junctions (*Figure 1A*; third column; *Figure 1—figure supplement 1D*, respectively). Dimerization-defective ATL-1 or ATL-3 mutants, carrying the R217Q or R213Q mutations (*Byrnes and Sondermann, 2011*), respectively, distributed throughout the ER (*Figure 1A*, fourth and fifth column). Thus, the punctate localization of ATL-3 to three-way junctions is due to its dimerization with endogenous ATL. The different localization patterns of ATL-1 and ATL-3 have been attributed to the fact that ATL-1 hydrolyzes GTP faster than ATL-3; faster GTP hydrolysis would lead to less concentration in junctions (*Yan et al., 2015*).

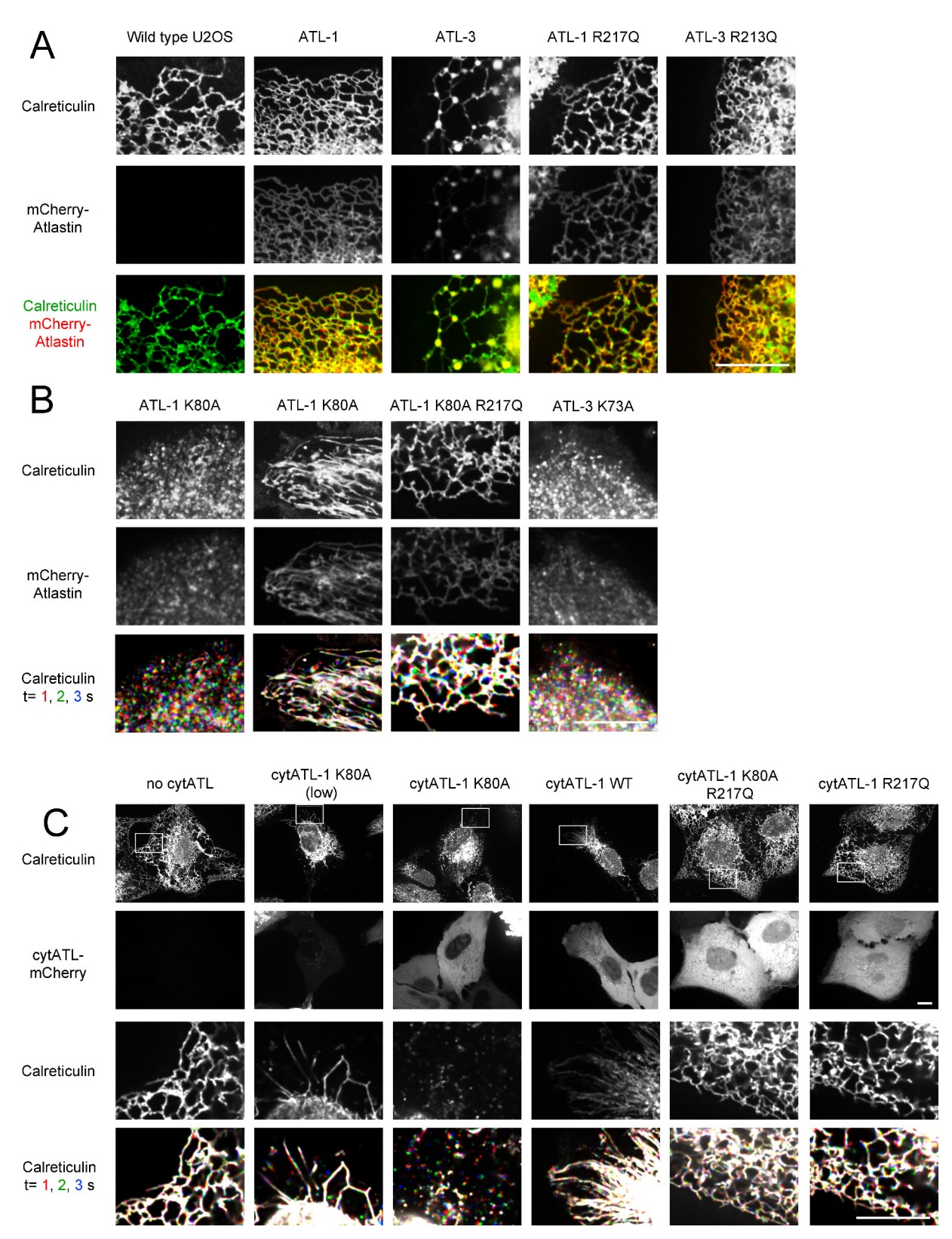

**Figure 1.** ATL is required to maintain tubules and junctions in mammalian cells. (**A**) Peripheral ER network of U2OS cells expressing GFP-calreticulin from the endogenous promoter in wild type cells. Where indicated, the cells also stably expressed mCherry-tagged wild type ATL-1 or ATL-3 or the

*Figure 1 continued on next page*

*Figure 1 continued*

corresponding dimerization-defective mutants (ATL-1 R217Q or ATL-3 R213Q). Scale bar = 10 µm. (**B**) As in (**A**), but with cells stably expressing mCherry-tagged ATL-1 or ATL-3 mutants defective in GTP hydrolysis (ATL-1 K80A or ATL-3 K73A). The first and second columns show ATL-1 K80A-expressing cells, in which the ER is either fragmented or converted into long, unbranched tubules. ATL-1 K80A R217Q is both GTPase- and dimerization- defective. The bottom row shows three time points for calreticulin staining. Stationary pixels appear white, while those moving appear in colors. Scale bar = 10 µm. (**C**) As in (**A**), but with cells stably expressing mCherry-tagged cytoplasmic fragments of wild type ATL-1 (cytATL-1 WT), a mutant defective in GTP hydrolysis (cytATL-1 K80A) at low or high level, or a mutant defective in dimerization (cytATL-1 R217Q). CytATL-1 K80A R217Q contains both mutations. The bottom two rows show a close-up image of the peripheral network of the cells depicted in the first two rows. The bottom row shows three time points for calreticulin staining. Stationary pixels appear white, while those moving appear in colors. Scale bar = 10 µm.

The following figure supplement is available for figure 1:

**Figure supplement 1.** Effect of ATL overexpression on ER morphology in U2OS cells.

Next we expressed dominant-negative, GTPase-defective mutants of fluorescently-tagged ATL. Expression of ATL-1 K80A caused the conversion of most of the ER into small vesicles, which rapidly moved around the cell, as shown by time-lapse imaging (*Figure 1B*; first column; the lowest row shows three different time points with different colors; see also *Video 1*). In many cells, long, unbranched tubules were also seen (*Figure 1B*; second column; *Video 2*). These membrane tubules were aligned with microtubules and seemed to be pulled out from a membrane reservoir near the nucleus after cell division (not shown). Indeed, treatment of the cells with the microtubule-disassembling drug nocodazole led to the disappearance of the long membrane tubules (*Figure 1—figure supplement 1E*). Expression of a double mutant of ATL-1, which is both GTPase- and dimerization-defective (ATL-1 K80A R217Q) left the reticular ER intact (*Figure 1B*; third column), demonstrating that the ATL-1 K80A mutant causes ER morphology changes by forming inactive dimers with endogenous ATL. In agreement with previous experiments (*Rismanchi et al., 2008*), we find that ATL-1 R217Q causes ER morphology defects in cells that express high amounts of this protein, but this is not caused by interference with endogenous ATL, but rather by non-specific membrane crowding. Vesiculation of the ER was also observed with dominant-negative, GTPase-defective ATL-2 or ATL-3 mutants (ATL-3 K73A or ATL-2 K107A; *Figure 1B*; fourth column; *Figure 1—figure supplement 1F*, respectively).

Expression of a cytoplasmic fragment of the GTPase-defective, dominant-negative ATL-1 K80A mutant (cytATL-1-K80A-mCherry) converted the ER into long, unbranched tubules at low expression

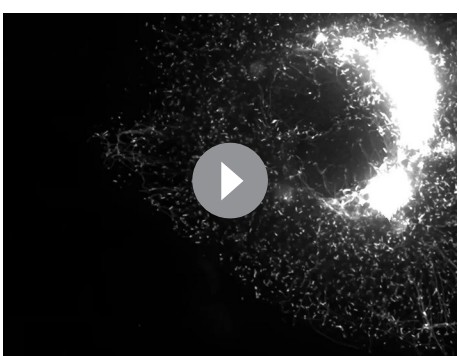

**Video 1.** ER in U2OS cells expressing dominant-negative ATL-1. U2OS cell stably expressing GFP-ATL-1 K80A (GTPase-defective mutant) with most of the peripheral ER fragmented. Images were acquired with a spinning disk confocal microscope at 0.05 sec intervals for 5 sec. The video is displayed at the same rate it was acquired (20 frames per sec). Image scale: 87 × 66 µm.

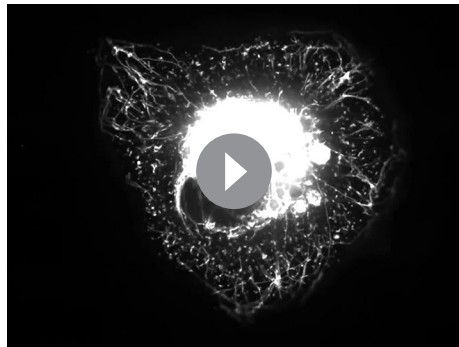

**Video 2.** ER in U2OS cells expressing dominant-negative ATL-1. U2OS cell stably expressing GFP-ATL-1 K80A (GTPase-defective mutant) with many unbranched tubules and a fragmented peripheral ER. Images were acquired with a spinning disk confocal microscope at 0.05 sec intervals for 5 sec. The video is displayed at the same rate it was acquired (20 frames per sec). This cell is also depicted in *Figure 1— figure supplement 1A*, left panel. Image scale: 87 × 66 µm.

levels (*Figure 1C*; second column), and into small vesicles at high levels (third column). A cytoplasmic fragment of wild type ATL-1 (cytATL-1-mCherry) only generated long, unbranched tubules, even at high expression levels (fourth column), likely because it interacts with endogenous ATL only transiently during the GTPase cycle and is therefore a weaker inhibitor than cytATL-1 K80A. Consistent with this assumption, cytATL-1 K80A localized more prominently to the ER than wild type cytATL-1 (not shown), and non-dimerizing versions of cytATL-1 K80A or cytATL-1, which carry the R217Q mutation (*Byrnes and Sondermann, 2011*), did not affect the morphology of the ER (columns five and six).

With the exception of a report in which cytATL was overexpressed (*Moss et al., 2011*), ATL inhibition was thought to generate only long, unbranched tubules, rather than cause ER-vesiculation. This is probably due to the use of fixed cells, in which small vesicles are lost or difficult to image. On the other hand, the phenotype is similar to that described for ATL-depleted *Drosophila* larvae (*Orso et al., 2009*). ER fragmentation has not been observed in *S. cerevisiae* cells lacking the ATL orthologue Sey1p (*Hu et al., 2009*), perhaps because ER SNAREs provide an alternative pathway of ER fusion in this organism (*Anwar et al., 2012*).

## ATL is required to maintain an ER network in *Xenopus* egg extracts

To test whether acute inhibition of ATL function affects the integrity of the ER network, we employed the *Xenopus* egg extract system. As reported previously, an ER network can be generated from membrane fragment in a crude interphase extract (*Figure 2A*); network formation is prevented when membrane fusion is inhibited by the addition of a dominant-negative cytoplasmic fragment of *Xenopus* ATL (cytATL) at the beginning of the reaction (*Wang et al., 2013*). *Xenopus* ATL is homologous to mammalian ATL-2 and is the only ATL annotated in the *Xenopus* genome. Surprisingly, we found that cytATL also disassembled a preformed network. At low concentrations, a fragmented tubular network remained (*Figure 2B*). At higher concentrations, only small membrane structures were observed (*Figure 2C*). A non-dimerizing mutant of cytATL (cytATL R232Q) had no effect on the integrity of the network (*Figure 2D*), likely because it did not bind to the endogenous ATL. Inhibition of endogenous ATL by addition of GTPγS also disassembled the network (*Figure 2E*). Similar results were obtained with a fractionated system, consisting either of *Xenopus* membranes and membrane-depleted cytosol (*Figure 2F*) or of membranes only (*Figure 2—figure supplement 1A*). These results indicate that ATL function is not only required to form an ER network, but also to maintain it.

ATL was also required to maintain an ER network in mitotic (meiotic) extracts. A dimerization-defective cytATL-R232Q mutant did not affect the mitotic ER network; as in untreated mitotic extracts (*Wang et al., 2013*), many small sheets connected by short tubules were seen (*Figure 2G*). In contrast, blocking the function of endogenous ATL by addition of cytATL resulted in the disassembly of the network (*Figure 2H*). Similar results were obtained with a mitotic network that was generated by incubating isolated membranes with cytosol that was moved into mitosis by the addition of non-degradable cyclin (*Figure 2—figure supplement 1B*).

To follow the disassembly of the ER network in real time and minimize its mechanical disruption, we used a microfluidics device. A network was first generated in a flow-chamber from a crude extract, and then cytosol, containing cytATL fused to GFP (cytATL-GFP), was slowly introduced. When cytATL-GFP arrived in the chamber, the network disassembled, starting at the site where cytATL-GFP arrived (*Figure 3*; *Video 3*). Close observation of the disassembly reaction showed that tubules frequently first detach from one another (arrows) before they crumble and convert into smaller structures (*Figure 3*). The initial detachment of the tubules suggests that some three-way junctions contained tethered, rather than fused tubules, and that ATL inactivation caused their untethering.

The final structures of network disassembly were larger than the small vesicles that serve as starting material for network formation. Occasionally, dynamic sheets were seen from which short tubules emerge and retract (not shown). These structures are possibly explained by residual ATL activity. They were much more abundant during early time points of network formation (*Figure 4A*), or if network formation was slowed down by partial inhibition of ATL, for example, by adding a low concentration of GTP to a membrane-only reaction (*Figure 4B*), or by adding GDP and AlF$_4^-$ to a reaction consisting of membranes and cytosol (*Figure 4C*). These intermediates in network formation are dynamic, small structures from which tubules continuously emanate and retract, although the disappearance of some tubules may simply be due to their bending or movement out of the focal plane

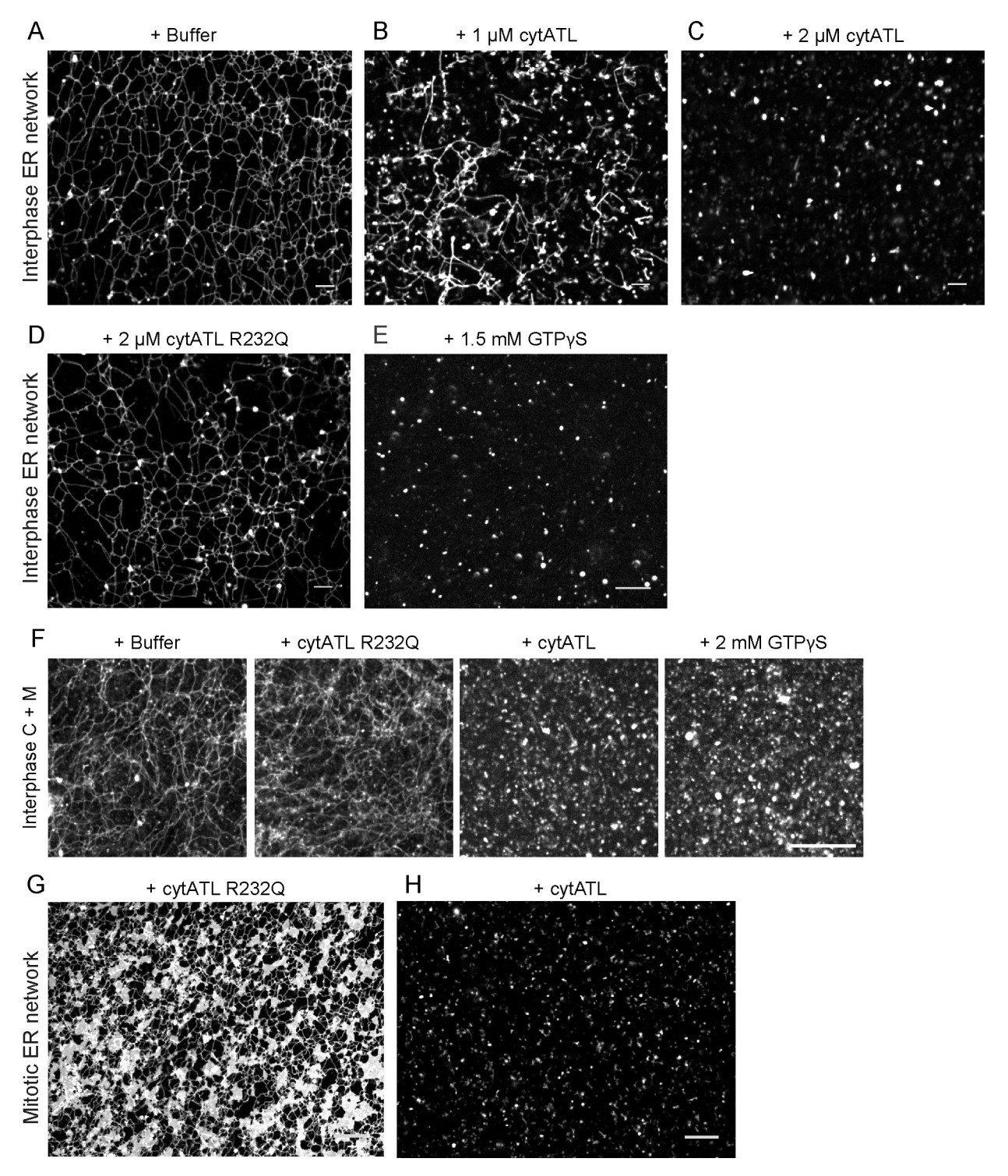

**Figure 2.** ATL is required to maintain an ER network in *Xenopus* egg extracts. (**A**) An ER network was generated with a crude interphase *Xenopus* egg extract and stained with the lipophilic fluorescent dye $DiIC_{18}$. The sample was imaged with a spinning disk confocal microscope. Scale bar = 10 µm. (**B**) As in (**A**), but in the presence of 1 µM cytoplasmic fragment of *Xenopus* ATL (cytATL). Scale bar = 10 µm. (**C**) As in (**A**), but with 2 µM cytATL. Scale bar = 10 µm. (**D**) As in (**A**), but with 2 µM of the dimerization-defective mutant fragment cytATL R232Q. Scale bar = 10 µm. (**E**) As in (**A**), but with 1.5 mM GTPγS. Scale bar = 10 µm. (**F**) An ER network was generated with interphase cytosol, light membranes, and an energy-regenerating system. After 30 min, buffer, 2 µM cytATL, 2 µM cytATL R232Q, or 2 mM GTPγS were added. The membranes were stained with octadecyl rhodamine. Scale bar = 20

*Figure 2 continued on next page*

*Figure 2 continued*

µm. (**G**) A mitotic ER network was generated with a crude *Xenopus* extract containing DiIC$_{18}$ and 2 µM cytATL R232Q. Scale bar = 10 µm. (**H**) As in (**G**), but with 2 µM wild type cytATL. Scale bar = 10 µm.

The following figure supplement is available for figure 2:

**Figure supplement 1.** ER network formed with *Xenopus* extract fractions is disassembled by ATL inactivation.

(*Videos 4*, *5*). These results indicate that during network formation, ATL first mediates the fusion of small vesicles into small, dynamic structures, which subsequently give rise to longer tubules. Inactivation of ATL disassembles the tubules into small membrane structures. Our experiments also indicate that, at endogenous levels, the reticulons and DP1 are insufficient to maintain the integrity of tubules in the absence of ATL function.

Functional ATL localized mostly to three-way junctions in an ER network assembled from crude *Xenopus* extracts, as demonstrated by the addition of fluorescently labeled, affinity-purified antibodies raised against *Xenopus* ATL (*Figure 5A*). In contrast, when cytATL labeled with the fluorescent dye Alexa 488 was added at low concentrations, the label was seen throughout the tubules (*Figure 5B*). These molecules likely represent inactive dimers formed with endogenous ATL; the fraction of inactivated endogenous ATL is apparently small enough to maintain the integrity of the network. Similar results were obtained with a fractionated system, consisting of membranes and cytosol. Again, antibodies to *Xenopus* ATL stained three-way junctions (*Figure 5—figure supplement 1A*), whereas cytATL-GFP stained the entire network (*Figure 5—figure supplement 1B*; left panel). As expected, cytATL R232Q-GFP, which fails to bind endogenous ATL, localized to the cytosol (*Figure 5—figure supplement 1B*; rightmost panel). These results indicate that functional ATL localizes to three-way junctions, whereas inactivated ATL distributes throughout the tubules.

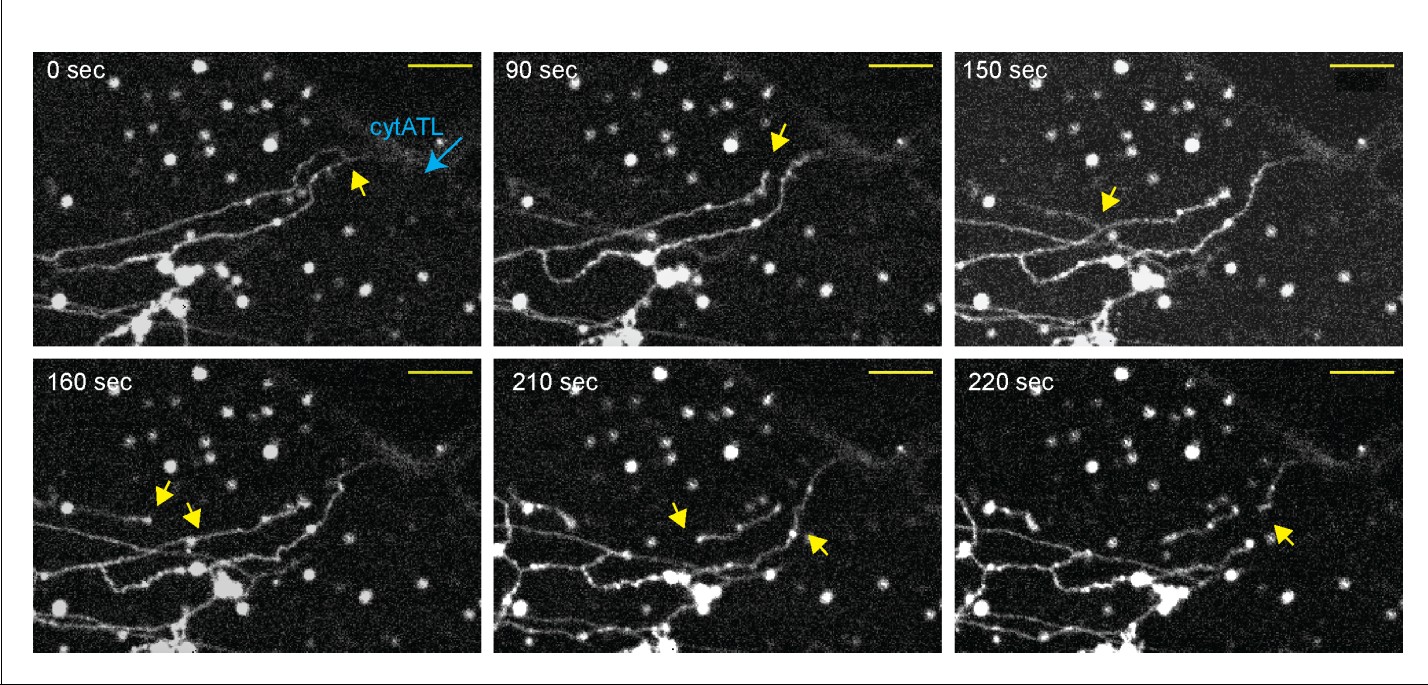

**Figure 3.** Real-time disassembly of an ER network in the presence of cytATL. A network was formed from a crude *Xenopus* egg extract in the presence of the dye DiIC$_{18}$ in a computer-controlled microfluidics device. CytATL-GFP (5 µM) in cytosol containing an energy regenerating system was then slowly perfused into the chamber at a total laminar flow rate of 0.5 µL/min from multiple ports (one is indicated by a blue arrow). The arrival of cytATL-GFP arrival in the chamber was monitored by GFP fluorescence (not shown). Images represent snapshots of a real-time video (see *Video 3*). Yellow arrows point to the detachment or breakage of tubules. Scale bar = 10 µm.

We noticed that inhibition of ATL function affected network formation more than network maintenance. The addition of low concentrations of GDP plus $BeF_3^-$, a mixture that locks endogenous ATL into an inactive, dimeric state, prevented the *de novo* formation of an ER network, but left a preformed network intact (*Figure 6*). At higher concentrations, both network formation and maintenance were affected. Similar observations were made with GTPγS (*Figure 6—figure supplement 1*). These experiments show that the *de novo* formation of an ER network is more sensitive to ATL inhibition than the maintenance of a preformed network.

## A balance is needed between ATL and Rtn

Next we tested the effect of the reticulons on ER morphology, using live-cell imaging of stable mammalian cell lines. We first expressed in U2OS cells a mCherry-fusion of Rtn4a, which has a long cytoplasmic N-terminus preceding the conserved membrane-embedded, reticulon-homology (RHD) domain (*Voeltz et al., 2006*). At low expression levels, mCherry-Rtn4a distributed together with GFP-calreticulin throughout the tubules of the ER (*Figure 7A*). At higher levels, unbranched tubules were seen (*Figure 7B*). Often these tubules were depleted of GFP-calreticulin, suggesting that these were too narrow to accommodate luminal proteins, as observed previously in transient overexpression experiments (*Hu et al., 2008*). In addition, some of the ER was fragmented; the membrane fragments contained both GFP-calreticulin and mCherry-Rtn4 and moved rapidly (*Figure 7C*; *Video 6*), but they seemed to be larger than in cells expressing dominant-negative ATL mutants (*Figure 1B*). Perhaps, they correspond to fractured tubules. Fragmentation of the ER by overexpression of Rtn4a had not been observed in fixed cells, again highlighting the need for live-cell imaging.

The co-overexpression of Rtn4a and wild type ATL-1 or ATL-3 restored the integrity of the network, making the membrane fragments disappear and returning tubular junctions (*Figure 7D*). This effect is due to the fusion activity of ATL because the co-expression of the dimerization-defective ATL-1 R217Q mutant did not restore the network (*Figure 7—figure supplement 1A*). As expected, the co-overexpression of the dominant-negative ATL1- K80A mutant and Rtn4a resulted in both long, unbranched tubules and small vesicles (*Figure 7E* and *Figure 7—figure supplement 1B*). Interestingly, Rtn4a and ATL seemed to segregate at least partially, perhaps because Rtn displaces ATL molecules from long tubules. Taken together, our results suggest that an appropriate balance between ATL and Rtn4a is required for network formation; too little ATL function, or too much Rtn4a, results in similar morphology defects, i.e. long, unbranched tubules and fragmented ER.

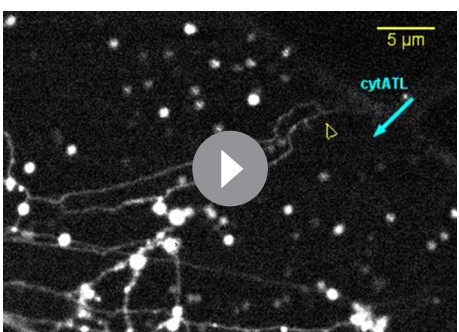

**Video 3.** Disassembly of an in vitro generated ER network by cytATL. An ER network was formed in a microfluidics chamber from *Xenopus* crude extract containing $DiIC_{18}$. *Xenopus* egg cytosol containing 5 µM cytATL-GFP was slowly perfused into the chamber using a computer-controlled pneumatic pump (blue arrow indicates perfusion port) while images were acquired with a spinning disk confocal microscope at 10-sec intervals for 9 min. The video is shown at 3 frames per sec. Yellow arrowheads indicate ER disassembly events. Still images of this video were used for *Figure 3*. Scale bar = 5 µm.

## Lnp determines the abundance of three-way tubular junctions in mammalian cells

Next we tested the role of lunapark (Lnp), arguably the least understood player in ER morphology. We generated a CRISPR knock-out of lunapark in U2OS cells that express GFP-calreticulin (*Figure 8—figure supplement 1A*). Cells lacking Lnp exhibited a proliferation of peripheral sheets and a reduction of tubules and junctions (*Figure 8A*; quantification in *Figure 8B*). However, the tubular network did not completely disappear in all cells. Immunostaining with reticulon antibodies showed that the residual tubular ER was often pushed to the periphery of the cells (*Figure 8—figure supplement 1B*). Thus, Lnp is not essential to form tubules or three-way junctions, but it affects their abundance. These results are consistent with the ER morphology reported for a Lnp knock-out in *S. cerevisiae* (*Chen et al., 2012*), although the small cell size of this organism makes the distinction between tubules and sheets difficult.

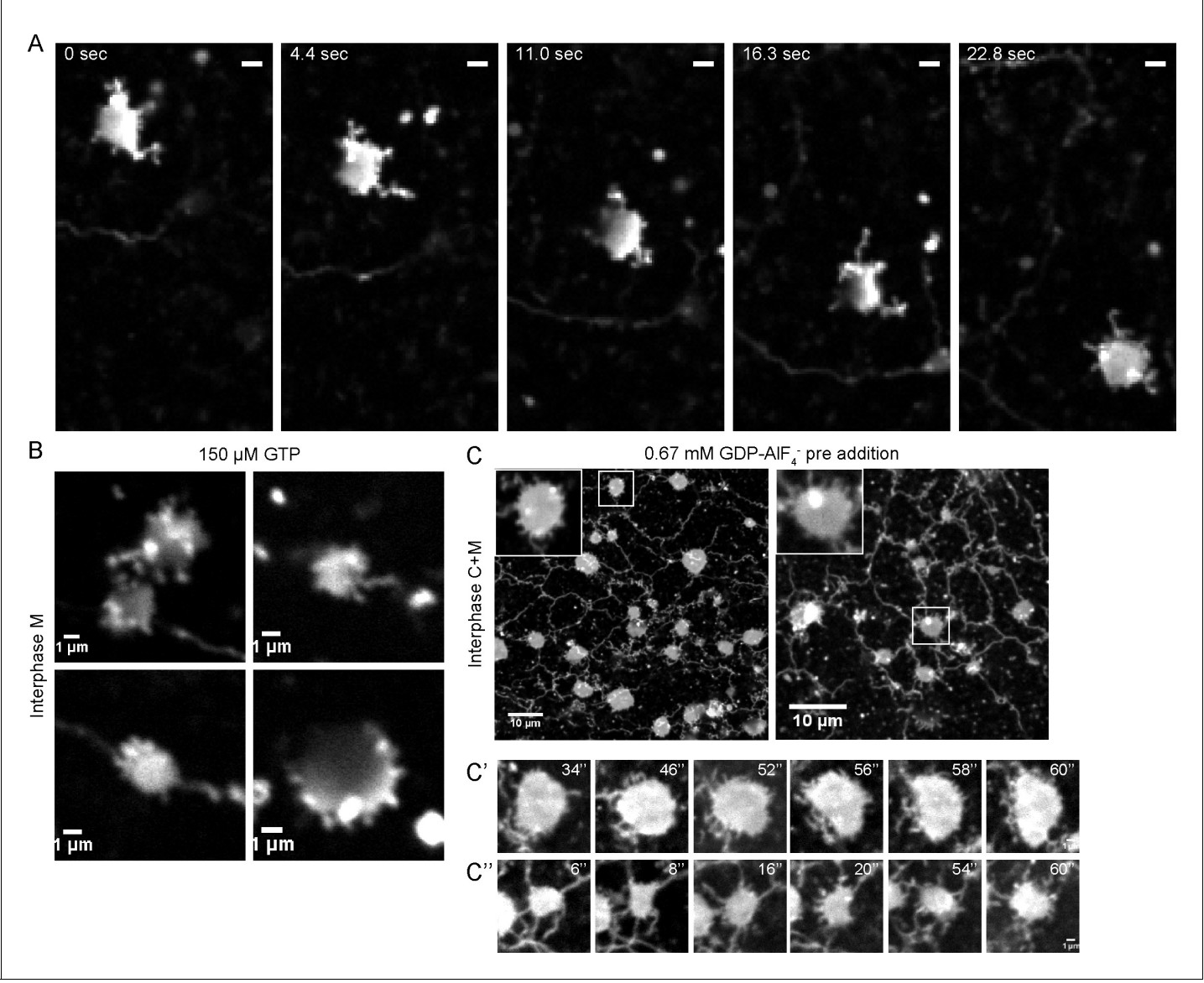

**Figure 4.** Intermediates during in vitro ER network formation. (**A**) DiIC$_{18}$-prelabeled light membranes were mixed with buffer and an energy regenerating system. The sample was imaged immediately by confocal microscopy. Scale bar = 2 μm. (**B**) *Xenopus* egg light membranes were incubated in the presence of 150 μM GTP and octadecyl rhodamine. Scale bar = 1 μm. (**C**) Interphase *Xenopus* egg cytosol, light membranes, 0.67 mM GDP-AlF$_4^-$, and an energy-regenerating system were incubated for 30 min. The membranes were stained with octadecyl rhodamine. Scale bars = 10 μm. Insets and images in **C'** and **C''** show magnified views of small sheets from which short tubules emanate. Scale bars = 1 μm.

When a fusion of wild type Lnp with mCherry (Lnp-mCherry) was stably expressed at low levels in CRISPR knock-out cells, the reticular network was restored; peripheral sheets were reduced in favor of tubules and junctions (*Figure 8C*; second column). Lnp-mCherry localized preferentially to punctae at three-way junctions (*Figure 8C*). At higher expression levels, Lnp-containing sheets became more prominent (*Figure 8D*), and at still higher levels, tubular structures with extremely large diameters were formed (>500 nm; *Figure 8D*; *Figure 8—figure supplement 1C*).

To better understand how Lnp is targeted to three-way junctions, we stably expressed various Lnp mutants at low levels in CRISPR knock-out cells. Lnp consists of an N-terminal myristoylated glycine (*Moriya et al., 2013*), a coiled-coil domain (CC1), two closely spaced trans-membrane segments (TM1 and TM2), another coiled-coil domain (CC2), a predicted unstructured segment, a Zn$^{2+}$-finger

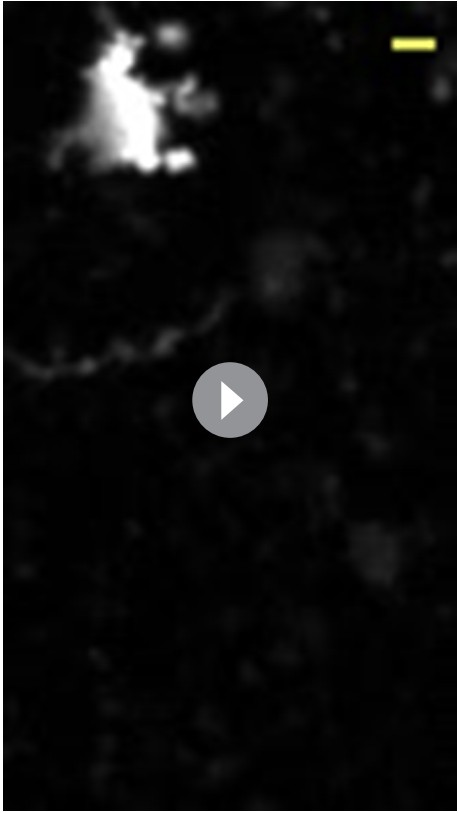

**Video 4.** Intermediates of an in vitro generated ER network at early time points. An ER network assembly reaction was prepared by mixing DiIC$_{18}$-prelabeled *Xenopus* egg light membranes with buffer and an energy regenerating system. The sample was imaged immediately using a spinning-disk confocal microscope at 0.5-sec time intervals for 39 sec. The video is shown at 10 frames per sec. Scale bar = 2 μm.

domain implicated Lnp dimerization (*Casey et al., 2015*), and a C-terminal unstructured segment (*Figure 9A*). A Lnp mutant, in which the N-terminal myristoylation site was abolished (G2A mutation), failed to localize to junctions (*Figure 9B*). Likewise, when a short, unrelated linker sequence was inserted after CC1 (*Figure 9B*) or before CC2 (not shown), Lnp no longer concentrated in punctae (*Figure 9B*), suggesting that the two coiled-coil domains need to be in register. Finally, truncation of Lnp before the Zn$^{2+}$-finger domain also abolished Lnp localization to three-way junctions, whereas a truncation at a more C-terminal site had no effect (*Figure 9B*). In all cases, localization of Lnp to three-way junctions correlated with the conversion of peripheral sheets into a tubular network, indicating that Lnp exerts its effect on ER morphology at junctions. At high levels, all Lnp mutants generated sheets and aberrant structures (not shown). Taken together, these results show that localization to three-way junctions requires several Lnp features, including the Zn$^{2+}$-finger implicated in dimerization. Indeed, pull-down experiments with extracts of mammalian cells expressing different Lnp constructs show that the region containing the Zn$^{2+}$-finger is required for Lnp-Lnp interaction (*Figure 9—figure supplement 1A,B*).

## Interplay of ATL and Lnp

To test whether Lnp function is dependent on ATL, we generated stable cell lines that overexpress both proteins as fluorescently tagged constructs. The large sheets observed when Lnp-mCherry was expressed at high level in wild type U2OS cells disappeared when wild type GFP-ATL-3 or GFP-ATL-1 were co-expressed (*Figure 10A*; compare second and third with first column). The ER network was restored, and the ATL proteins localized as in wild type cells. GFP-ATL-3 was always close to Lnp-mCherry, and was often found at the edges of the Lnp-containing structures, but the two proteins did not co-localize (*Figure 10A*; second column). Expression of GFP-ATL-1 also caused Lnp-mCherry to redistribute from large to small sheet-like structures (*Figure 10A*; third column), even though ATL-1 itself localized mostly throughout the tubules. Taken together, these results indicate that ATL affects the localization of Lnp, although the bulk of ATL and Lnp do not seem to interact physically. In agreement with this conclusion, no interaction was seen in pull-down experiments (not shown). Our experiments also suggest that ATL functions upstream of Lnp, a conclusion that is supported by the observation that Lnp overexpression did not significantly affect the localization of the ATLs (*Figure 10A*), and did not reverse ER fragmentation in cells

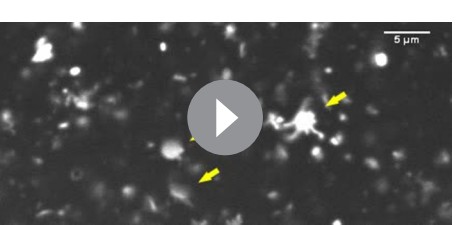

**Video 5.** Intermediates of an in vitro generated ER network at early time points. As in *Video 4*, but the image was acquired at 0.2-sec time intervals for 5 sec. Scale bar = 5 μm. The video is shown at 5 frames per sec.

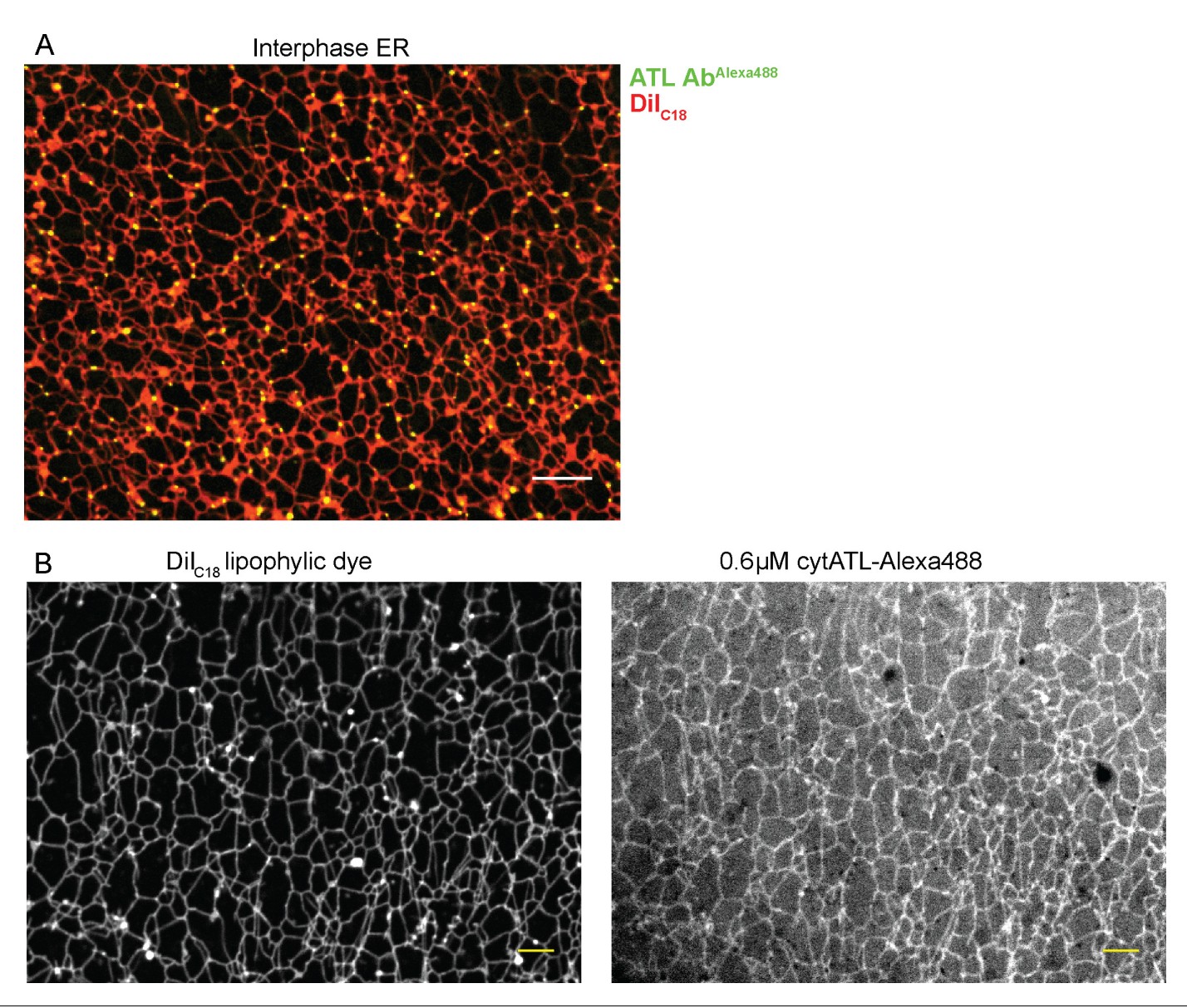

**Figure 5.** Localization of ATL in an in vitro generated ER network. (**A**) An interphase ER network was generated with a crude *Xenopus* egg extract containing the dye DiIC$_{18}$. Endogenous ATL was visualized by including 16 nM Alexa488-labeled, affinity-purified antibodies raised against *Xenopus* ATL (ATL Ab$^{Alexa488}$). Note that ATL localizes preferentially to three-way junctions. Scale bar = 10 μm. (**B**) An interphase ER network was assembled as in (**A**) and labeled with DiIC$_{18}$ and 0.6 μM Alexa488-labeled cytATL. Note that cytATL localizes throughout the tubules, marking the position of inactivated endogenous ATL. Scale bar = 10 μm.

The following figure supplement is available for figure 5:

**Figure supplement 1.** Localization of endogenous and inactivated ATL in a *Xenopus* ER network.

expressing dominant-negative ATL mutants (*Figure 10—figure supplement 1A*).

## ATL and Rtn can bypass the function of Lnp

The stable expression of wild type ATL in Lnp-lacking cells converted peripheral ER sheets into tubules and junctions (*Figure 10B*; second and third column), whereas dominant-negative ATL mutants fragmented the ER or led to long, unbranched tubules (*Figure 10—figure supplement 1B*).

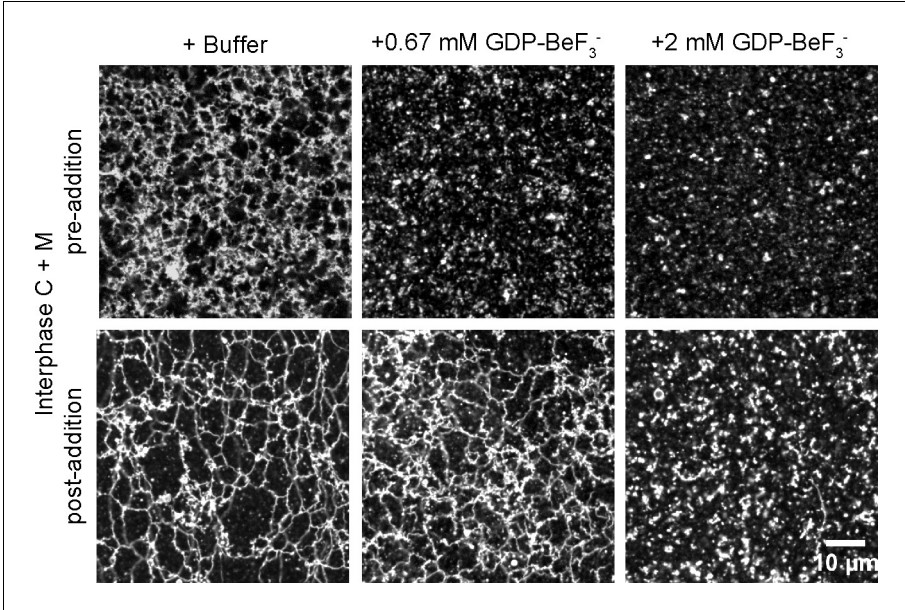

**Figure 6.** Effect of GTP analogs on ER network formation and maintenance. GDP-BeF$_3^-$ was added at 0.67 mM or 2 mM either before (upper panels) or after (lower panels) formation of an ER network from *Xenopus* egg cytosol (**C**), light membranes (**M**), and an energy regenerating system. The membranes were stained with octadecyl rhodamine. Scale bar = 10 μm.

The following figure supplement is available for figure 6:

**Figure supplement 1.** ER network maintenance is less sensitive to ATL inhibition than ER network formation.

These are the same phenotypes observed when these constructs were expressed in wild type cells (*Figure 1*). Similarly, as in wild type cells, low-level expression of Rtn4a in Lnp knock-out cells resulted in restoration of the tubular network (*Figure 10B*; fourth column), whereas at higher levels it caused the appearance of long tubules and membrane fragments (not shown). These results show that, at elevated levels, ATL and Rtn do not depend on Lnp function.

## Lnp is required to form three-way tubular junctions in *Xenopus* extracts

To further investigate the role of Lnp in shaping the ER, we used the *Xenopus* extract system. We reasoned that a cytoplasmic fragment of *Xenopus* Lnp (cytLnp), containing the sequence following TM2 (*Figure 11A*), might interact with the endogenous protein and serve as a dominant-negative reagent. Indeed, when purified cytLnp was added to a crude extract at the beginning of a network formation reaction, small sheets appeared over time, which replaced three-way tubular junctions (*Figure 11B;* the reduction of three-way junctions is quantitated in *Figure 11C*). In addition, the tubules of the network became shorter. The number of three-way junctions was also reduced when cytLnp was added to a network formation reaction performed with membranes and cytosol (*Figure 11D*) or with membranes alone (*Figure 11—figure supplement 1A*). In these cases, the junctions were converted into bright membrane structures, which look different from those observed in crude extracts (*Figure 11B*). Whether these structures are sheets therefore remains uncertain. Three-way junctions were also abolished when cytLnp was added after network formation (*Figure 11—figure supplement 1B*). Affinity-purified antibodies raised against *Xenopus* Lnp had the same effect (*Figure 11E* and *Figure 11—figure supplement 1C*), confirming that Lnp inactivation causes the change in ER morphology. A C-terminal fragment of cytLnp (cytLnp-C) had the same effect on the integrity of three-way junctions as cytLnp, whereas two different N-terminal fragments (cytLnp-N1 and –N2) were inactive (*Figure 11F*). Pull-down experiments confirmed that cytLnp and cytLnp-C, but not cytLnp-N1 or –N2, interact with endogenous Lnp (*Figure 11—figure supplement 1D,E*). The Zn$^{2+}$-finger-containing constructs cytLnp and cytLnp-C likely form inactive oligomers with

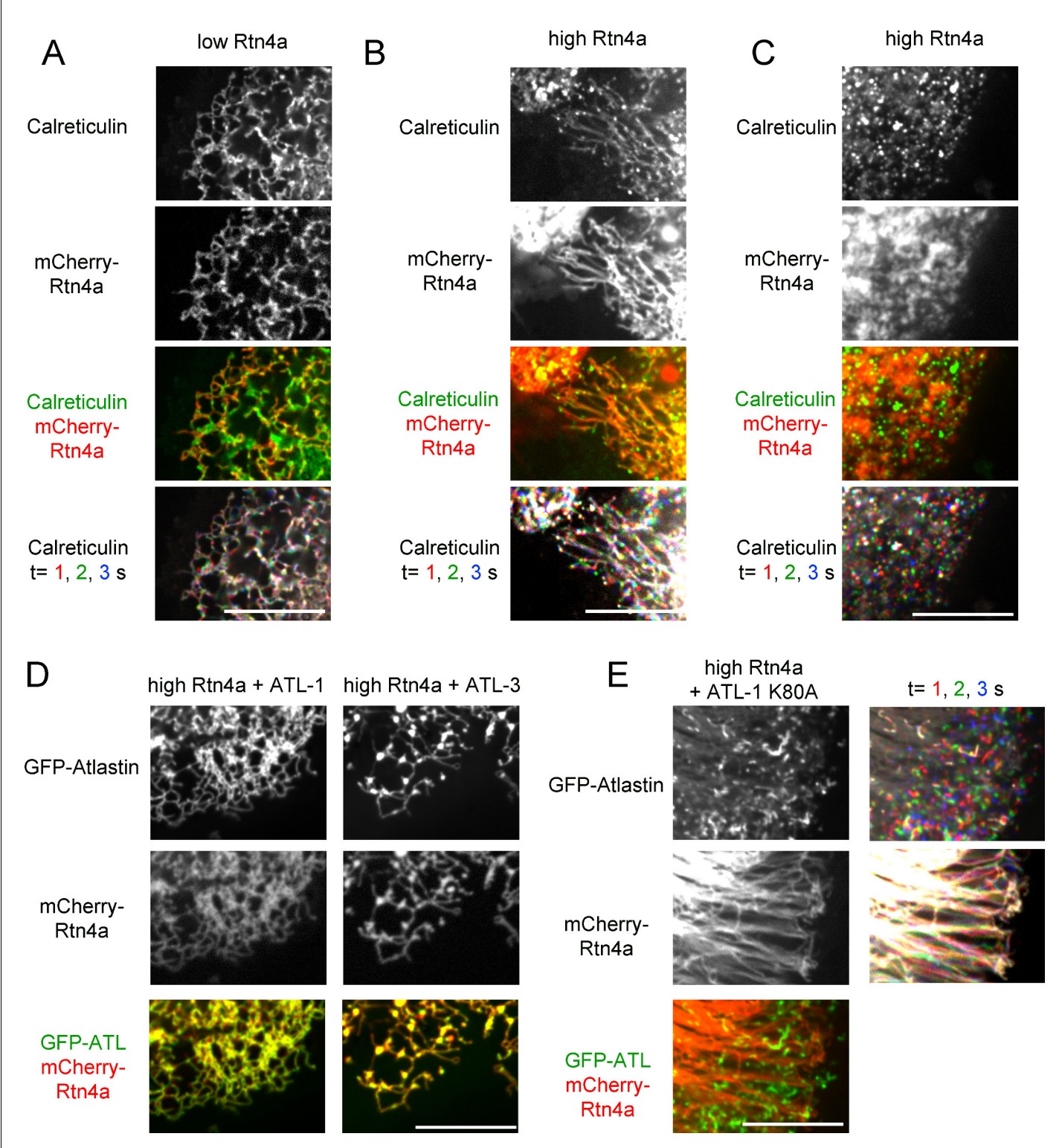

**Figure 7.** Interplay between ATL and the reticulons. (**A**) Peripheral ER network in a U2OS cell expressing GFP-calreticulin under the endogenous promoter in wild type cells or in cells stably expressing mCherry-tagged Rtn4a. The bottom row shows three time points. Stationary pixels appear white, while those moving appear in colors. Scale bar = 10 µm. (**B**) As in (**A**), but with a higher expression level of mCherry-Rtn4a, resulting in unbranched tubules. (**C**) As in (**B**), but showing a cell with fragmented ER. (**D**) As in (**B**), but with cells that also stably express GFP-ATL-1 or GFP-ATL-3.
*Figure 7 continued on next page*

*Figure 7 continued*

(E) As in (B), but with cells that also express a GTPase defective ATL-1 mutant (GFP-ATL-1 K80A). The second column shows three time points for both the ATL and Rtn channels. Stationary pixels appear white, while those moving appear in colors. Scale bar = 10 μm.

The following figure supplement is available for figure 7:

**Figure supplement 1.** Co-overexpression of ATL mutants and Rtn4a in U2OS cells.

the endogenous full-length protein. Consistent with this assumption, gel filtration experiments indicated that cytLnp and cytLnp-C form oligomers, in contrast to cytLnp-N1 and cytLnp-N2, which elute as monomers (*Figure 11—figure supplement 1F*). Taken together, these results are consistent with those obtained in mammalian cells, and show that Lnp stabilizes three-way tubular junctions and tubules. Lnp-Lnp interactions involving the $Zn^{2+}$-finger containing domain seem to be required for the function of Lnp.

## Mitotic phosphorylation of Lnp

The network formed with crude *Xenopus* extracts after addition of cytLnp looks indistinguishable from that in mitosis, consisting of small sheets connected by multiple, short tubules (compare *Figures 11B* and *2G*). In both cases, the number of three-way junctions was reduced to about the same extent compared with an interphase extract (*Figure 11C*). Similar results were obtained with a fractionated system consisting of membranes and cytosol; here too, a mitotic network (*Wang et al., 2013*) was indistinguishable from an interphase network formed in the presence of cytLnp (*Figure 11D*). The similarity suggests that Lnp is inactivated during mitosis.

We tested whether Lnp is phosphorylated during mitosis, as this may provide a mechanism of inactivation. Lnp is indeed phosphorylated in mitotic (meiotic) *Xenopus* extracts. Endogenous Lnp, which ran as a single band in interphase extracts, was converted into two higher molecular weight species (*Figure 12A*). The same size shifts were seen with His-tagged cytLnp added to either interphase or mitotic extracts (*Figure 12B*). Both higher molecular weight species disappeared when cytLnp was pulled down with cobalt beads, indicating that Lnp modification is reversible. Addition of phosphatase inhibitors during pull-down resulted in the disappearance of the higher molecular weight species, but preserved the lower one (*Figure 12C*). Addition of N-ethylmaleimide preserved both bands. These results indicate that at least the smaller of the two modified species corresponds to mitotically phosphorylated Lnp. It is possible that the larger species carries an additional modification. Mass spectrometry revealed that *Xenopus* Lnp is mitotically phosphorylated at multiple sites, most of which are located in the unstructured domain, located between CC2 and the $Zn^{2+}$-finger domain (*Figure 12D*). It is unclear whether all sites are equally modified in a given Lnp molecule.

Lnp is also mitotically phosphorylated in mammalian U2OS cells. Immunoblotting showed that endogenous Lnp increases in size and decreases in abundance in cells arrested in mitosis with the microtubule-depolymerizing drug nocodazole (*Figure 12E*). Like other proteins regulated during the cell cycle in mammalian cells, Lnp seems to be degraded following phosphorylation. Overexpressed Lnp-mCherry also underwent a size shift during mitosis and slightly increased in abundance when its degradation was prevented by addition of the proteasome inhibitor bortezomib (*Figure 12F*). A Lnp-mCherry mutant in which phosphorylation sites were mutated to Ala did not undergo a size shift in mitosis nor a change in abundance with bortezomib treatment. The mitotic phosphorylation sites in mammalian Lnp were in the same domain as in *Xenopus* Lnp (*Figure 12D*).

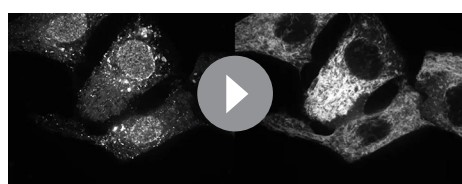

**Video 6.** ER morphology in U2OS cells after overexpression of Rtn4a. U2OS cells expressing GFP-calreticulin under the endogenous promoter as well as stably expressing mCherry- Rtn4a. Left panel: calreticulin signal, right panel: mCherry-Rtn4a signal. Images were acquired with a spinning disk confocal microscope at 0.4 sec intervals for 4 sec. The video is displayed at the same rate it was acquired (2.5 frames per sec). Image scale: 87 × 66 μm for each panel.

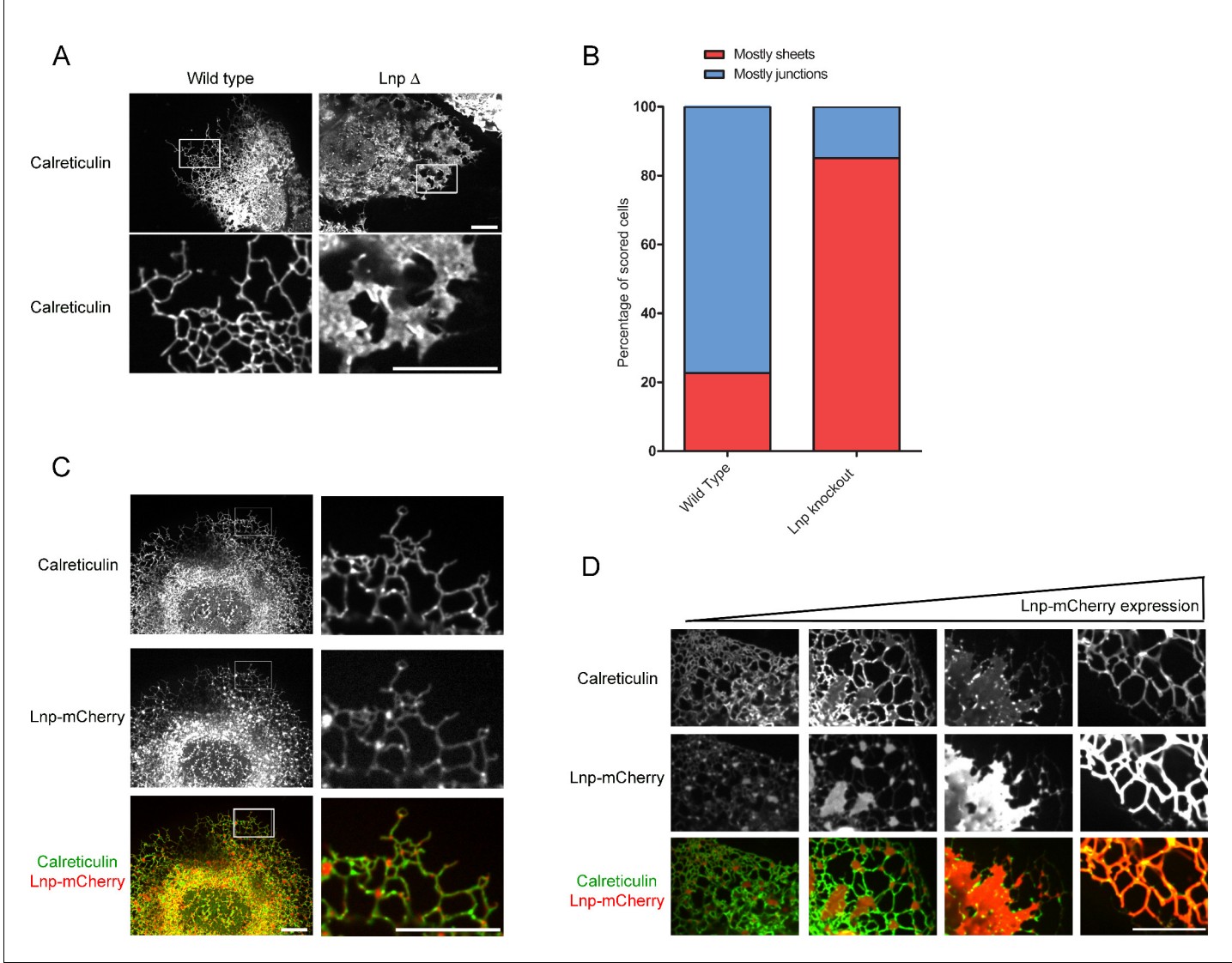

**Figure 8.** Lnp determines the abundance of three-way tubular junctions in mammalian cells. (A) Views of wild type U2OS and Lnp-deleted (LnpΔ) cells expressing GFP-calreticulin from the endogenous promoter. LnpΔ cells were generated by CRISPR targeting the start codon of the *LNP* gene. The bottom row shows magnifications of the boxed areas of the peripheral ER. Scale bars = 10 µm. (B) Quantification of the *LNP* deletion phenotype depicted in A. Wild type and LnpΔ cells were scored blindly for the appearance of peripheral ER. (C) Peripheral ER in a LnpΔ cell expressing GFP-calreticulin, as well as stably expressing a low level of Lnp-mCherry. Scale bars = 10 µm. (D) As in (C), but with cells expressing increasing levels of Lnp-mCherry (left to right). Scale bars = 10 µm.

The following figure supplement is available for figure 8:

**Figure supplement 1.** ER morphology in U2OS cells lacking or overexpressing Lnp.

Pull-down experiments showed that the oligomerization of cytLnp is weakened in mitotic *Xenopus* extracts (*Figure 12—figure supplement 1A*). A phosphomimetic cytLnp mutant containing several Ser/Thr to Glu mutations (cytLnp-E) also had a weakened tendency to oligomerize (*Figure 12—figure supplement 1B*). These results suggest that mitotic phosphorylation reduces the interaction of Lnp molecules, which in turn leads to the destabilization of three-way junctions and tubules. Consistent with this model, a phosphomimetic mutant of full-length Lnp did not localize to three-way junctions when expressed at low levels in CRISPR knock-out cells, in contrast to a mutant in which the same residues were changed to Ala (*Figure 12—figure supplement 1C,D*). Thus, mitotic

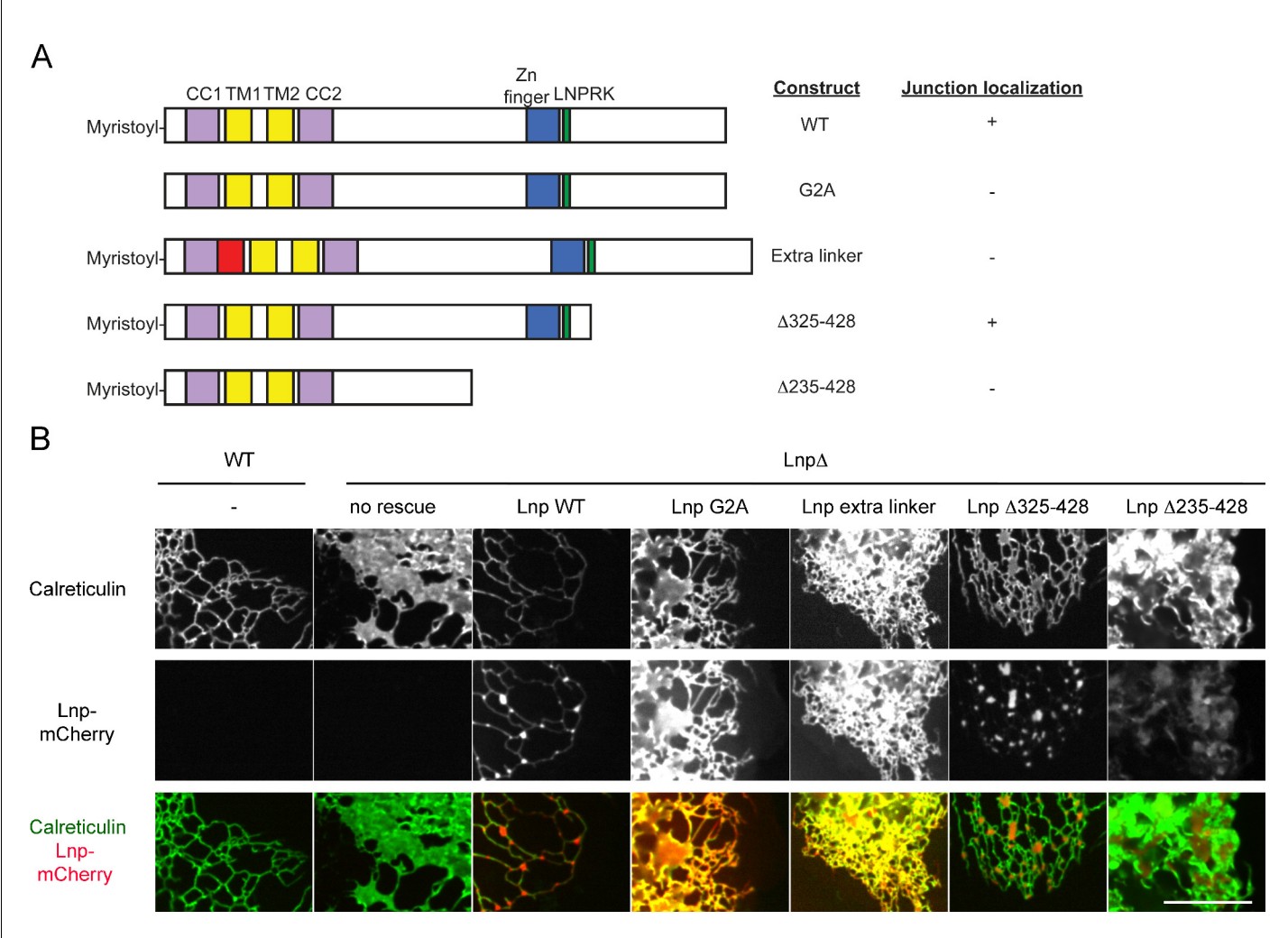

**Figure 9.** Lnp domains important for junction localization. (**A**) Schematic representation of wild type (WT) Lnp and mutants tested for proper localization in the peripheral ER. CC1, CC2, coiled-coil domains 1 and 2, respectively; TM1, TM2, trans-membrane segments 1 and 2, respectively; LNPRK, lunapark motif (single letter code); the segment indicated in red was inserted (extra linker). (**B**) Peripheral ER in U2OS cells expressing GFP-calreticulin under the endogenous promoter in wild type or Lnp-lacking cells (LnpΔ). Where indicated, the cells also stably expressed wild type or mutant Lnp at low levels. Scale bar = 10 μm.

The following figure supplement is available for figure 9:

**Figure supplement 1.** Lnp-Lnp interaction is mediated by the $Zn^{2+}$-finger domain-containing region.

phosphorylation may cause the disappearance of Lnp from three-way junctions; the junctions are then converted into larger sheets.

## Discussion

Our results provide insight into how the ER-shaping proteins ATL, Rtn, and Lnp cooperate to generate and maintain a tubular ER network. We show that ATL is not only required to form a network, but also has a hitherto unrecognized role in maintaining it. When ATL is inhibited, three-way tubular junctions disappear and much of the network disassembles into small membrane structures, indicating that physiological concentrations of the curvature-stabilizing proteins Rtn and DP1 are insufficient to maintain the integrity of tubules in the absence of ATL function. Overexpression of Rtn4a leads to long, unbranched tubules and fragmented ER. Both phenotypes are restored when ATL is

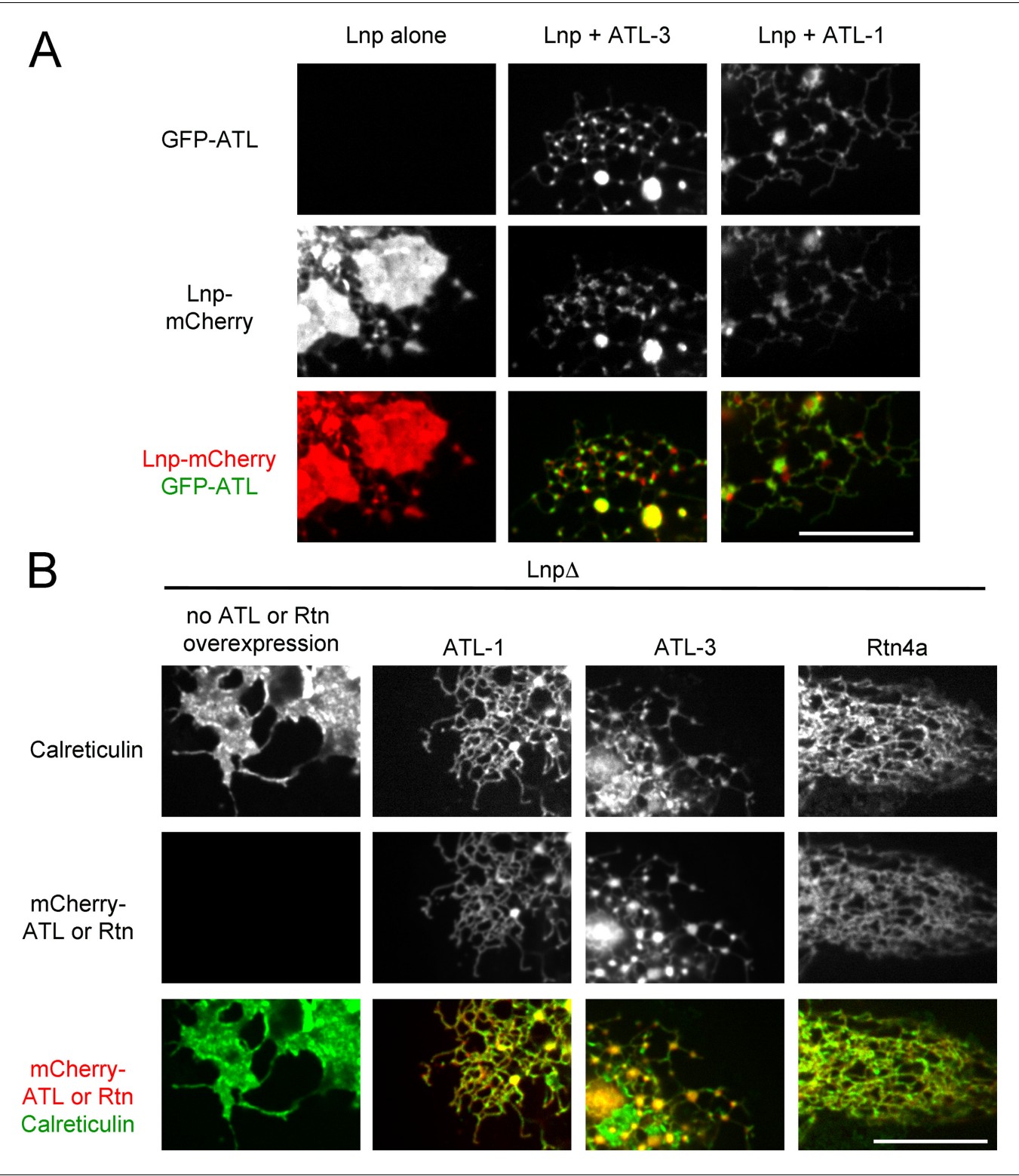

**Figure 10.** Interplay of ATL, Lnp, and Rtn. (**A**) Peripheral ER in U2OS cells stably expressing Lnp-mCherry alone or together with GFP-ATL-3 or GFP-ATL-1. Scale bar = 10 micron. (**B**) Peripheral ER in Lnp-lacking U2OS cells expressing GFP-calreticulin. Where indicated, the cells also stably expressed mCherry-tagged ATLs or Rtn4a. Scale bar = 10 micron.

The following figure supplement is available for figure 10:

*Figure 10 continued on next page*

*Figure 10 continued*

**Figure supplement 1.** ATL acts upstream of Lnp in U2OS cells.

co-overexpressed. Taken together, our data indicate that a balance between ATL and Rtn function is required. Lnp is not essential for network formation or maintenance; its inactivation or deletion only reduces the abundance of three-way junctions and tubules in favor of larger sheets. Lnp is phosphorylated during mitosis, which weakens its oligomerization and may contribute to the characteristic tubule-to-sheet conversion of the ER during mitosis.

Previous experiments have established that ATL mediates the fusion of ER membranes (*Orso et al., 2009*; *Bian et al., 2011*). A GTP-hydrolysis- dependent conformational change in ATL trans-dimers pulls the apposing membranes together so that they can fuse. ATL-mediated fusion requires multiple ATL molecules in each membrane (*Liu et al., 2015*), and multiple cycles of GTP hydrolysis (*Liu et al., 2015*; *Saini et al., 2014*). It is therefore likely that ATL molecules are concentrated at the site of fusion between two ER tubules. Cis-dimerization in the same membrane (*Liu et al., 2015*) and nucleotide-independent interactions through the TM segments (*Liu et al., 2012*) may also increase ATL localization at these sites before and after fusion. Eventually, however, ATL would diffuse away from three way junctions and distribute throughout the tubules. This model may explain why mammalian ATL-2 and ATL-3, as well as *Xenopus* ATL, preferentially localize to three-way junctions. Mammalian ATL-1 is only marginally concentrated, perhaps because fewer molecules are required for a successful fusion event or because it diffuses away more rapidly from the fused junction.

ATL-mediated fusion of tubules is clearly required to form and maintain an ER network. When ATL function is moderately inhibited, long, unbranched tubules are formed, likely because these cannot be linked into a network. When ATL is more strongly inhibited, even tubules are disrupted. A possible explanation for the essential role of ATL in maintaining the integrity of tubules is based on the observation that free ends of tubules are rarely seen in mammalian cells or in networks generated in vitro with *Xenopus* egg extracts; essentially all tubule ends are anchored either in three-way junctions or are associated with molecular motors or microtubule tips. We therefore propose that tubules are unstable when they have free ends. Free ends have a high membrane curvature at the tip, and the elastic energy would be reduced if they retracted in favor of other tubules that are anchored, or if they shed small vesicles. Shedding of vesicles from a free tubule end would be favored by Rtn, which itself imposes high curvature on the membrane. At physiological concentrations, Rtn would stabilize the high membrane curvature of tubules in cross-section, but it may prefer to sit in a vesicle of the same radius, particularly if it formed a hydrophobic insertion, rather than a scaffold around the bilayer. ATL would be essential for tubule integrity, as it would counteract the vesicle shedding at the tips of tubules by capturing shedding vesicles via its tethering function and then fusing them back onto the tubule. This model is consistent with our observations of early intermediates during network formation in *Xenopus* extracts. Initially, vesicles seem to fuse into dynamic small membrane structures that continuously emanate and retract short tubules. These tubules have free ends and are unstable, and their dynamic nature may correspond to a situation in which their shortening is reversed by their ATL-mediated fusion with small vesicles. Our interpretation implies that the shedding and fusion of vesicles at tubule ends occur at comparable rates, whereas the breakage of tubules or three-way junctions may be kinetically disfavored. Indeed, live-cell imaging of the ER in mammalian cells or *Xenopus* egg extracts shows that tubules rarely undergo fission in the middle of the tubule. Instead, retraction of tubules is a frequent event and might involve the shedding of vesicles at tubule tips.

In addition to ensuring the integrity of the free ends of tubules, ATL might stabilize the ER network at three way junctions. Because ATL needs multiple cycles of trans-dimerization to achieve fusion, it may tether tubules together at three way junctions that have not yet fused. ATL inactivation would cause the disassembly of a preformed ER network by the untethering of these tubules, leading to the appearance of free, unstable tubule ends. It is unclear whether this model can explain the disassembly of the entire network in the *Xenopus* system, as it would imply that a high percentage of three-way junctions is actually only tethered, rather than fused. It is also possible that ATL is

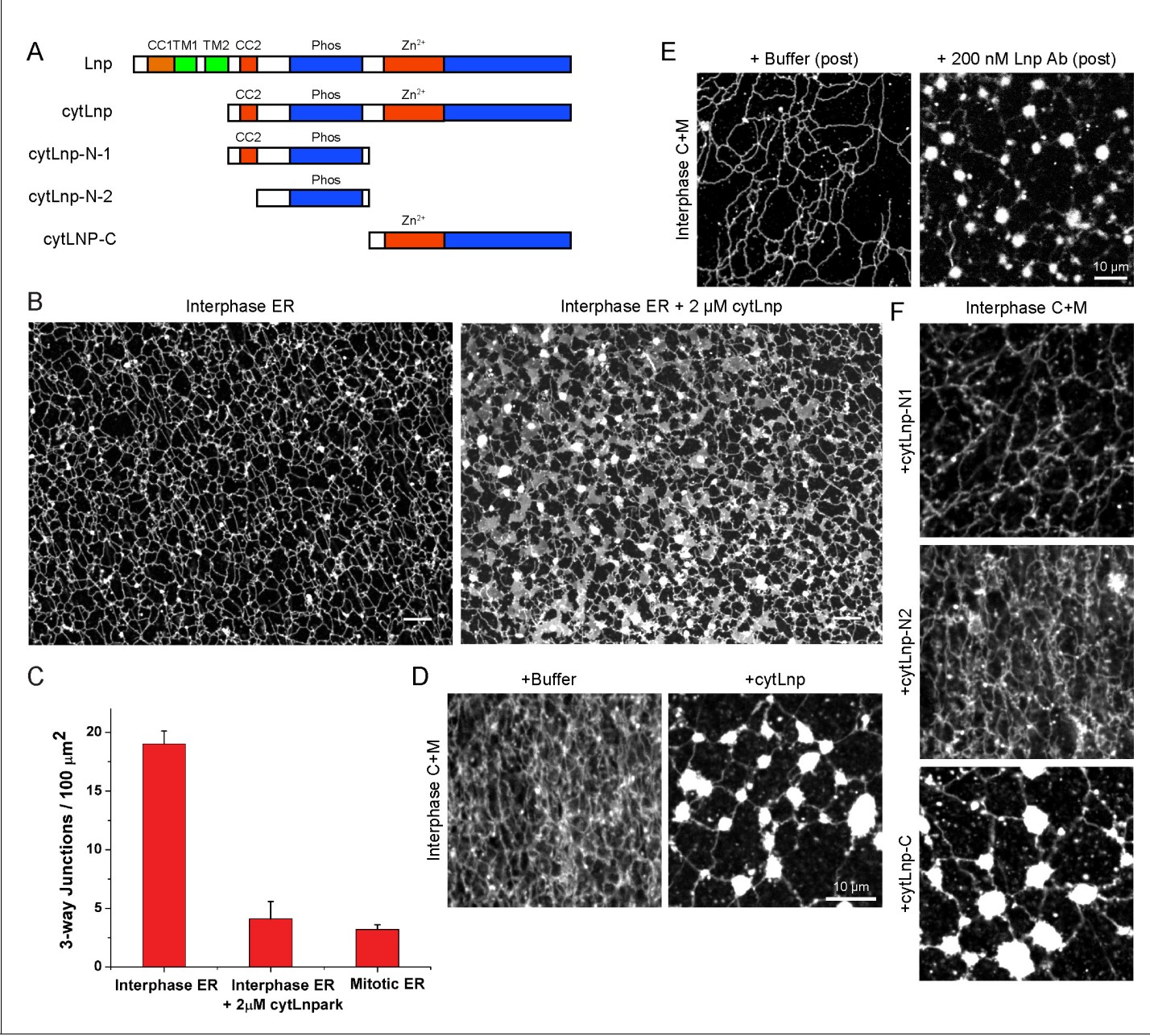

**Figure 11.** Effect of Lnp inactivation on an in vitro generated ER network. (**A**) Schematic representation of wild type and mutant *Xenopus* Lnp. Phos indicates the domain phosphorylated during mitosis. (**B**) An ER network was generated for 20 min with crude *Xenopus* egg extract in the presence of the dye DiIC$_{18}$ and either buffer (left panel) or 2 µM of a cytoplasmic fragment of Lnp (cytLnp). Scale bar = 10 µm. (**C**) The number of three-way junctions at different conditions was quantitated. Error bars indicate the mean ± SD of three independent experiments. (**D**) An interphase ER network was formed with *Xenopus* egg cytosol (C), light membranes (M), an energy regenerating system, in the presence or absence of 5 µM cytLnp. The network was stained with octadecyl rhodamine. Scale bar = 10 µm. (**E**) Buffer or 200 nM of affinity-purified Lnp antibodies were added to a preformed interphase network generated with cytosol, membranes, and an energy regenerating system. Scale bar = 10 µm. (**F**) As in (**D**), except that 5 µM cytLnp-N1, cytLnp-N2 or cytLnp-C were added at the beginning of the network formation reaction. Scale bars = 10 µm.
The following figure supplement is available for figure 11:

**Figure supplement 1.** Cytoplasmic fragments of Lnp interfere with the function of endogenous *Xenopus* Lnp.

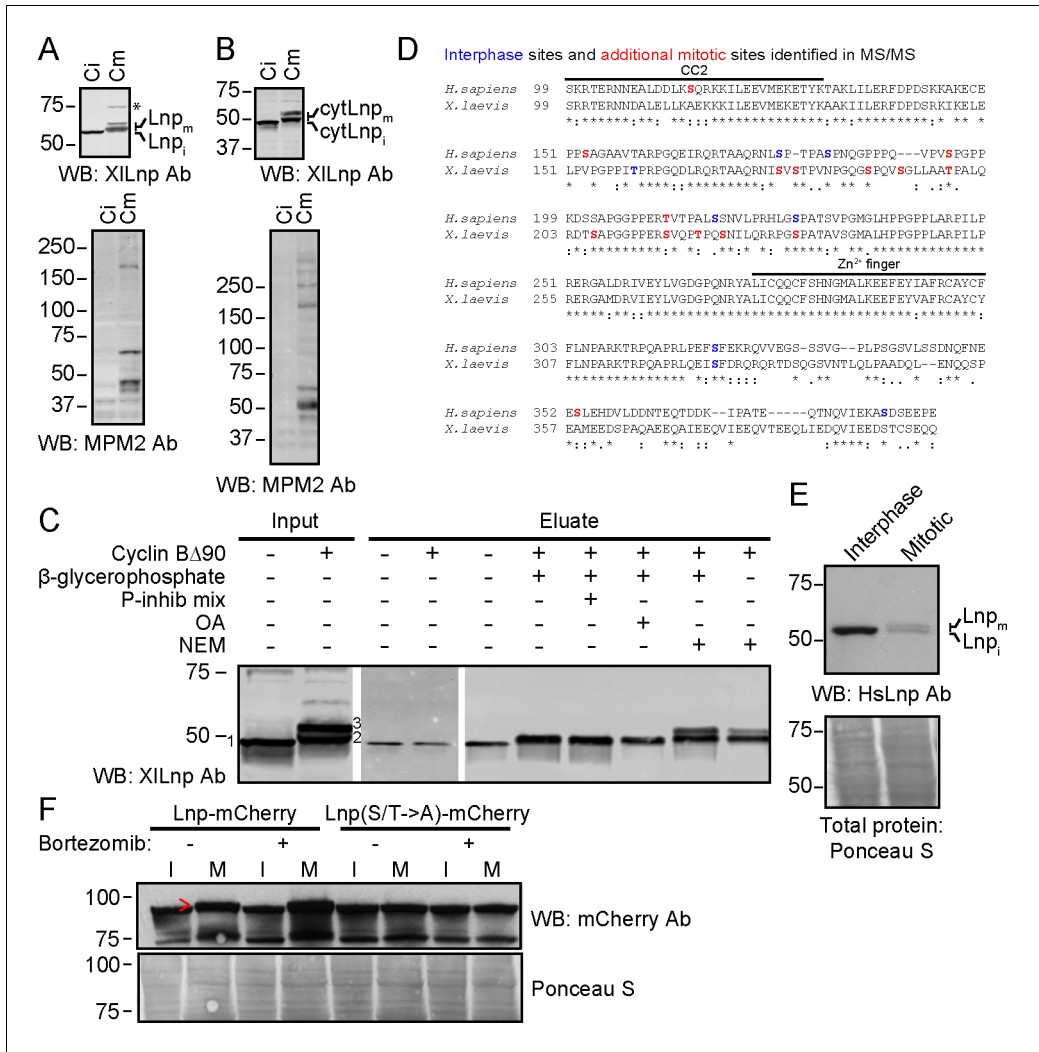

**Figure 12.** Mitotic phosphorylation of Lnp. (**A**) *Xenopus* egg interphase cytosol, membranes, and an energy regenerating system were incubated with buffer (C$_i$) or non-degradable cyclin BΔ90 (C$_m$) for 40 min. The samples were analyzed by SDS-PAGE and immunoblotting with X*enopus* Lnp antibodies (XlLnp Ab). The lower panel shows an immunoblot with MPM2 antibodies that recognize mitotically phosphorylated proteins. (**B**) As in (**A**), but cytLNP was incubated with cytosol in the absence of membranes. (**C**) Purified HA- and His-tagged cytLnp (HA-cytLnp) was incubated with interphase cytosol with or without cyclin BΔ90. An aliquot was analyzed directly by SDS-PAGE (input), while the remainder was incubated with cobalt resin in the presence of various inhibitors (P-inhib is a phosphatase inhibitor cocktail (Sigma); OA, okadaic acid; NEM, N-ethylmaleimide). Material eluted from the beads with imidazole (eluate) was analyzed by SDS-PAGE and immunoblotting with *Xenopus* Lnp antibodies. Band 1, unmodified HA-cytLnp; bands 2 and 3, mitotically modified Lnp. (**D**) Alignment of the cytosolic domains of human and *Xenopus* LNP sequences. Residues in blue are phosphorylated in interphase, as determined by mass spectrometry. Residues in red are additionally phosphorylated during mitosis. (**E**) U2OS cells were grown in complete medium and left untreated or treated with 100 nM nocodazole overnight. Interphase cells were scraped off, while mitotically arrested cells were collected as non-adherent cells. Equal amounts of total protein were analyzed by SDS-PAGE and immunoblotting with antibodies to human Lnp (HsLnp). Lnp$_i$ and Lnp$_m$, interphase and mitotic Lnp, respectively. (**F**) Lnp-mCherry or Lnp-mCherry with Ser and Thr phosphorylation sites mutated to Ala (S/T ->A) were stably expressed in U2OS cells. Cells were arrested in mitosis by incubation with 100 nM nocodazole overnight. Where indicated, 100 nM bortezomib was present overnight. Equal amounts of total protein from interphase (I) or mitotic (M) cells were analyzed by SDS-PAGE and immunoblotting with mCherry antibodies. The arrowhead indicates the position of mitotically phosphorylated Lnp.

The following figure supplement is available for figure 12:

**Figure supplement 1.** Mitotic phosphorylation weakens Lnp-Lnp interaction.

required to maintain the integrity of fused three-way junctions, for example, by stabilizing the negatively curved edge line bordering the triangular junctional sheet. We also found that a significantly stronger inhibition of ATL is required to disassemble a preformed network than to prevent its *de novo* formation. Perhaps, trans-dimerization and fusion are more sensitive to ATL inhibition than ATL concentration at three-way junctions. It has been proposed that the yeast ortholog of ATL, Sey1p, is merely a tethering molecule akin to Rab GTPases, and that the actual fusion is mediated by ER SNAREs (*Lee et al., 2015*). However, mutations in Sey1p and ER SNAREs have synthetic effects in *S. cerevisiae* (*Anwar et al., 2012*), indicating that they function in parallel pathways. Our present results, and those demonstrating that ATL and Sey1p mediate fusion on their own (*Orso et al., 2009*; *Bian et al., 2011*; *Anwar et al., 2012*), argue that tethering and fusion are just different stages of a reaction carried out by the same protein.

The proposed balance between ATL and Rtn function may explain our overexpression results. The overexpression of Rtn generates very narrow tubules, from which the luminal protein calreticulin is displaced. Assuming that ATL is also displaced, the appearance of long, unbranched tubules may simply be explained by their inability to form new three-way junctions or to fuse vesicles to the end of tubules. In addition, Rtn overexpression could lead to its oligomerization, which could contribute to the formation and stabilization of long tubules. Rtn4a overexpression also generates membrane fragments, which appear to be larger than those observed upon ATL inactivation. Perhaps, these fragments originate from the breakage of the narrow tubules. Co-overexpression of ATL with Rtn would rescue both phenotypes, unbranched tubules and fragmentation, as it allows ATL to localize into the tubules to restore three-way junction formation and to fuse membrane fragments together. Our studies were restricted to Rtn4a, and it remains to be seen whether other Rtn isoforms and DP1 behave in a similar manner.

Our results also provide some insight into the function of Lnp. This protein is clearly not essential for the generation or maintenance of an ER network. In mammalian cells, some reticular network is retained in its absence, and even in *Xenopus* extracts, where Lnp inactivation converts most three-way junctions into larger sheets, there is still a network. Many of these sheets are bordered by negatively curved edge lines, suggesting that Lnp is not a major stabilizer of such edges (*Shemesh et al., 2014*). Based on the observation that Lnp forms sheet-like structures when overexpressed in mammalian cells, it seems possible that Lnp localizes to the sheets of tubular junctions. This is consistent with the fact that ATL-3 sits at the edges of expanded junctional sheets in Lnp-overexpressing cells. Because ATL is absolutely required to generate tubular junctions, one may assume that ATL first forms a native junction and sits at its edges, and then Lnp moves to the interior sheet. Perhaps, a Lnp scaffold prevents the reticulons and DP1 from moving into the three-way junction and thus counteracts junction expansion; when Lnp is absent or inactivated, the three-way junction area could increase into a larger sheet. Alternatively, Lnp may somehow promote the curvature-stabilizing function of the reticulons and DP1, thus helping in tubule formation.

The localization of Lnp to three-way junctions requires its oligomerization. One domain that is needed for its localization and mediates dimerization is the $Zn^{2+}$-finger (our results and [*Casey et al., 2015*]). However, the coiled-coil domains are probably also important. These regions immediately flank the TMs in all species and our experiments suggest that their correct registry is required for Lnp's localization to three-way junctions. The N-terminal myristic acid is also important, as previously reported (*Moriya et al., 2013*), perhaps because it anchors the N-terminus in the membrane. How exactly the different features of Lnp cooperate to localize Lnp to three-way junctions remains to be clarified.

Our results confirm that the ER in mammalian cells and *Xenopus* extracts undergoes a tubule-to-sheet transition during mitosis (*Lu et al., 2009*; *Wang et al., 2013*). The ER in mitotic Xenopus extracts looks strikingly similar to that generated by Lnp inactivation, and the absence of Lnp in mammalian cells also causes a tubule-to-sheet conversion. These results suggest that Lnp inactivation may be responsible for, or at least contribute to, the ER morphology changes during mitosis. Consistent with this idea, Lnp is mitotically phosphorylated in *Xenopus* extracts and mammalian tissue culture cells. In mammalian cells, Lnp is also degraded, as seen with other mitotically phosphorylated proteins. Phosphorylation of Lnp interferes with its oligomerization, and phosphomimetic Lnp mutants no longer localizes to three-way junctions and have a lower propensity to form oligomers. The effect on oligomerization can be explained by the fact that most phosphorylation sites are located in a region between the second coiled-coil and $Zn^{2+}$ finger domains, both of which may be

involved in Lnp-Lnp interactions. Taken together, these results suggest that mitotic phosphorylation reduces Lnp's oligomerization, which in turn results in its departure from three-way junctions. The absorption of membrane lipids into the junctions would result in their expansion into larger sheets. It is possible that other factors, such as the disruption of the ER – cytoskeleton interaction, or the release of translating ribosomes from the ER, contribute to the transition from tubules to sheets during mitosis. Sheet formation during mitosis might be required to accommodate membrane proteins that prefer low membrane curvature areas and sit during interphase in the relatively flat nuclear envelope.

Taken together, our results suggest a simple model for the generation and maintenance of a tubular ER network. ATL is required to both form and maintain a network. It mediates the fusion of tubules by forming GTP hydrolysis-dependent trans-dimers and transiently sits in the newly formed three-way junctions. Lnp subsequently moves into the junctional sheets, a process that may require its oligomerization. The reticulons stabilize high membrane curvature in tubules and may cause vesicle shedding at free tubule ends, a process counteracted by ATL through its tethering and fusion activity. During mitosis, Lnp is phosphorylated, causing it to dissociate into monomers and leave the three-way junctions. This would result in the expansion of three-way junctions into larger sheets. Experiments with reconstituted, purified ER shaping proteins are required to test this simple model and to understand the exact molecular mechanism of network formation.

## Materials and methods

### Plasmids

Maltose-binding protein-cyclin BΔ90 (MBP-cyclin BΔ90) lacking the N-terminal destruction box of cyclin B was a gift from Dr Randy King (Harvard Medical School, MA). Codon-optimized cytATL (residues 1–462) of *Xenopus laevis* ATL was cloned into pGEX-4T-3 (GE Healthcare) as described previously (*Wang et al, 2013*). A fusion of cytATL and superfolder GFP was cloned in the pPROEX-HTb vector (Lifetechnologies). A cytosolic fragment of *Xenopus* lunapark (cytLnp) (99–441) and truncations thereof (cytLnp-N1 (99–238), cytLnp-N2 (124–238) and cytLnp-C (239–441)) were expressed in the pET28b vector as N-terminal His$_6$ fusions with or without a C-terminal HA tag. CytLnp was also cloned into a pET28b vector with or without a C-terminal SBP tag and an N-terminal 3C protease site after the His-tag (cytLnp-SBP), as well as with an N-terminal HA tag after the His-tag and a 3C protease site (HA-cytLnp) or with an N-terminal His$_{14}$ tag and a SUMO protease cleavage site (Sumo-cytLnp). A phosphomimetic cytLnp was generated, in which ten Ser/Thr sites modified only during mitosis (as determined by tandem mass spectrometry) were mutated to Glu (*Figure 12D*). This sequence was cloned into a pET28b vector expressing either a N-terminal His$_6$ and a C-terminal HA tag (cytLnp-E-HA), or an N-terminal His$_{14}$ tag and a SUMO protease cleavage site (Sumo-cytLnp-E). Point mutations were generated using the QuickChange Site Directed Mutagenesis Kit (Stratagene).

For mammalian gene constructs, Lnp, ATL, and Rtn genes were subcloned from plasmids described in previous papers (*Zhu et al., 2003*; *Voeltz et al., 2006*; *Shemesh et al., 2014*; *Rismanchi et al., 2008*). eGFP and mCherry tags were fused to proteins via Gibson or restriction cloning and inserts were ligated into pHAGE2 lentiviral construct using PacI and NotI sites.

Point mutations in the respective constructs were generated using the QuickChange Site Directed Mutagenesis Kit (Stratagene) or PCR. More extensive mutations (phosphomimetic, phosphoalanine variants of Lnp) were created by PCR-splicing of synthesized DNA fragments (Thermo Fisher).

CRISPR/Cas9-mediated genome editing was performed using the pX330 vector expressing wild type *S. pyogenes* Cas9 (Addgene) (*Cong et al., 2013*). The guide RNA (gRNA) used for Lnp knockout was GTGGATTATTTTCTCGATGG, targeting the start codon. gRNA for insertion of eGFP into the calreticulin gene was AGGAGCAGTTTCTGGACGG, targeting the end of the signal sequence. Endogenous calreticulin was tagged with GFP using a donor plasmid that inserted an eGFP and a tetra-alanine linker immediately after the signal peptide cleavage site of the CALR gene. Homology arms were 650 base pairs long on both sides. All plasmid inserts were confirmed by sequencing.

## Protein purification

All proteins were expressed in BL21-CodonPlus (DE3)-RIPL (Agilent) or BL21 *E. coli* strains (NEB). CytATL, cytATL(R232Q), and MBP-cyclin BΔ90 were purified, as described previously (*Wang et al., 2013*). CytATL-GFP, cytATL(R232Q)-GFP and all cytLnp proteins except cytLnp-SBP were isolated using a Ni-NTA resin (Thermo Scientific), followed by anion-exchange chromatograph (HiTrap Q HP, GE Healthcare) and gel filtration (Superdex 200, GE Healthcare). CytLnp-SBP was purified using a Ni-NTA resin, followed by purification on a streptavidin resin (Thermo Scientific) and gel filtration (Superdex 200, GE Healthcare). The proteins were snap-frozen in 20 mM HEPES pH 7.5, 150 mM KCl, 250 mM sucrose, and 1 mM dithiothreitol (DTT). For the gel filtration analysis in *Figure 11—figure supplement 1F*, 400 µg of purified His$_6$-cytLnp/-N1/-N2/-C in 200 µL were injected onto a Superdex 200 column in buffer containing 20 mM HEPES pH 7.5, 150 mM KCl, and 1 mM DTT.

## Affinity purification and labeling of antibodies

Antibodies to full-length *Xenopus* ATL and His$_6$-cytLnp were raised in rabbits. For affinity purification of antibodies, cytATL and cytLnp (5 mg) were desalted by gel filtration using phosphate saline buffer (PBS) pH 7.2. The protein was crosslinked to 2 mL Affigel-15 resin (Biorad) overnight at 4°C. Unbound protein was washed with elution buffer (100 mM glycine-HCl pH 2.5, 150 mM NaCl), and the column re-equilibrated with PBS buffer pH 7.2. To affinity-purify the antibodies, the resin was incubated with clarified crude serum for 4 hr at 4°C. The antibodies were eluted with 100 mM glycine HCl pH 2.5, 150 mM NaCl and immediately neutralized with 1 M Tris/HCl pH 8. The buffer was exchanged to 20 mM HEPES pH 7.5, 150 mM KCl, 250 mM sucrose by dialysis, and the sample was snap frozen.

ATL antibodies were labeled with Alexa 488-NHS ester (Invitrogen) as follows. CytATL-conjugated resin (0.1 mL) was incubated with 1 mL of crude clarified rabbit serum for 4 hr at 4°C. The resin was washed with labeling buffer (30 mM Hepes/KOH pH 8.4, 150 mM NaCl), and then incubated for 1 hr at room temperature with 1 mL labeling buffer containing 100 µg Alexa488-NHS ester. Unreacted dye was removed by extensive washing with labeling buffer, and labeled antibodies were eluted using acidic elution buffer, followed by immediate neutralization with 1 M Tris buffer pH 8.0. The buffer was then exchanged to 20 mM HEPES pH 7.5, 150 mM KCl, 250 mM sucrose using PD-10 columns (GE Healthcare) and snap frozen.

## Preparation of *Xenopus* Egg Extract

Metaphase-arrested crude *Xenopus laevis* egg extracts (CSF extract), interphase cytosol, light membranes, pre-labeled membranes, and de-membranated sperm were prepared as described (*Wang et al., 2013*).

## In vitro ER network formation with crude *Xenopus* egg extracts

ER network formation using crude extracts was performed as described previously (*Wang et al., 2013*). Briefly, a CSF extract was incubated with 100 µg/mL DiIC$_{18}$ for 45 min at 18°C. An interphase ER network was generated by adding 0.5 mM CaCl$_2$ to a fresh extract together with 1/10 of total volume of DiIC$_{18}$ pre-labeled CSF extract. The samples were incubated for 7 min at 18°C. A mitotic (meiotic) ER network was generated by omitting CaCl$_2$. The samples (8 µL) were placed between two PEG-passivated glass coverslips (22 × 22 mm) (passivation as previously described [*Wang et al., 2013*]), mounted on top of a metal slide with a 20 mm diameter hole, and sealed with VALAP (1:1:1 mix of vaseline, lanolin and paraffin). The mounted sample was incubated for 10 min at 16°C, and then allowed to equilibrate to room temperature for 5 min before imaging on a spinning-disk confocal microscope.

## In vitro ER network formation with cytosol and membranes

ER network formation with cytosol and membranes or membranes alone was performed as described previously (*Wang et al., 2013*). Briefly, cytosol and membranes were mixed at a 20:1 volume ratio, or membranes were mixed with ELB200 (50 mM HEPES/KOH, pH 7.5, 200 mM KCl, 2.5 mM MgCl$_2$, and 250 mM sucrose, 1 mM DTT) at a 1:20 ratio. All reactions contained an energy regenerating system, except for the experiment in *Figure 4B*, which was performed in the presence of 150 µM GTP. The samples were incubated for 15–30 min at room temperature. Proteins or

various GTP analogues were either added at the beginning of the network assembly or to a pre-formed network. An aliquot of the reaction was mixed with octadecyl rhodamine (Invitrogen) at 10 µg/mL and applied to a passivated No. 1.5 coverslip sandwich sealed with VALAP. When DiIC$_{18}$-pre-labeled interphase membranes were used, octadecyl rhodamine was omitted. In *Figure 4C*, the GDP/AlF$_4^-$ sample was generated by adding 3.35 mM MgCl$_2$, 6.7 mM NaF, 0.335 mM AlCl$_3$, and 0.67 mM GDP (final concentrations). In *Figure 6*, the GDP/BeF3$^-$ sample was generated by adding 3.35 mM MgCl$_2$, 5.36 mM NaF, 1.34 mM BeSO$_4$, and 0.67 mM GDP, or 10 mM MgCl$_2$, 16 mM NaF, 4 mM BeSO$_4$, and 2 mM GDP.

## ER network disassembly with a microfluidics device

The reactions were performed in 2 mm diameter open-top chambers of a commercial microfluidics plate (Cellasic Onix model M04L). Sample perfusion was software-controlled using pneumatic pumps (Cellasic Onix platform, EMD Millipore). The wells and conductive micro channels used in the experiment were previously drained of preserving buffer and flushed with *Xenopus* cytosol supplemented with an energy regeneration system to prevent dilution of the crude extract. The glass-bottom microfluidics plate was mounted on the stage of an inverted spinning-disk confocal microscope and imaged in real-time during the experiment. The 2 mm round chamber was rinsed twice with crude *Xenopus* egg extract. To assemble an interphase ER network in the chamber, 2 µL of extract pre-labeled with DiIC$_{18}$ was added, layered with 2 µL of silicone oil to prevent evaporation, and incubated for 15 min at room temperature. *Xenopus* cytosol supplemented with an energy regeneration mix and 5 µM GFP-cytATL was perfused from well #5 in the plate at 4 psi pump pressure (calculated flow of 0.5 µL/min according to manufacturer specifications). Arrival of GFP-cytATL at the ER network-containing chamber was followed by GFP fluorescence and took approximately 7 min.

## Mammalian cell line generation

U2OS cells were not authenticated by STR profiling. The cells were treated with plasmocin, but they may not be completely free of mycoplasma contamination. The cells were cultured in DMEM supplemented with 10% fetal bovine serum, 25 mM HEPES and penicillin/streptomycin. All stable cell lines were generated by lentiviral transduction using pHAGE2 vectors using standard methods. Briefly, lentivirus was packaged in 293T cells using Trans-IT (Mirus) to co-transfect packaging and lentiviral vectors. Six to twelve hours post-transfection, medium was refreshed, and virus-containing supernatants were collected for 24 hr, spun down, and added to U2OS cells at a 1:5 dilution. No blasticidin selection was used, in order to obtain populations with varying levels of gene expression. For fluorescent-protein-tagged viral constructs, cells were sometimes sorted by flow cytometry to obtain cells with certain levels of expression. For GFP insertions into the genomic *CALR* locus, cells were transfected with guide RNA encoding plasmid, donor DNA, and wild-type Cas9. GFP positive cells were selected several days post-transfection using flow cytometry. For *LNP* deletion, isolated clones were first screened by immunoblotting for absence of Lnp protein and then verified by PCR and sequencing of genomic DNA around the start codon.

## Antibodies and reagents used for experiments with mammalian cells

Primary antibodies used for mammalian cell experiments were: anti-calreticulin from rabbit (Abcam), anti-reticulon 4 from goat (Santa Cruz), anti-Lnp from rabbit (Sigma), HA (Roche), mCherry (polyclonal made in rabbits). Secondary antibodies for immunofluorescence were Alexa-conjugated donkey-antibodies directed against mouse, rabbit, or goat immunoglobulin (Invitrogen). Secondary antibodies for immunoblotting were HRP conjugated anti-goat (Thermo Fisher), aanti-rabbit and anti-mouse immunoglobulin (GE Healthcare). Anti-HA affinity matrix (Sigma) was used for pull-down experiments. Nocodazole was purchased from Sigma.

## Imaging mammalian cells

Live imaging of mammalian cells was performed inside an Okolab stage top incubator warmed to 37°C. Cells were grown in #1.5 glass bottom Mat-Tek dishes and switched to phenol red- free Opti-MEM just prior to imaging.

For immunostaining of fixed cells, U2OS were grown on acid-washed glass coverslips and fixed using 2% formaldehyde and 0.2% glutaraldehyde in PBS for 15 min, then washed and quenched

with 1 mg/mL sodium borohydride in PBS for 5 min. Cells were permeabilized by incubation in PBS with 0.1% Triton X-100 for 5 min. Primary and secondary antibodies were incubated with coverslips for 45 min each in blocking solution (1% low IgG-containing fetal bovine serum (FBS) in PBS). Coverslips were mounted with mounting medium (Vectashield) on slides.

## Fluorescence microscopy

All samples were visualized using a spinning disk confocal head (CSU-X1; Yokogawa Corporation of America) with Borealis modification (Spectral Applied Research) and a quad bandpass 405/491/561/642 dichroic mirror (Semrock). The confocal was mounted on a Ti inverted microscope (Nikon) equipped with a 60× Plan Apo NA 1.4 oil immersion objective or a 100x Plan Apo 100x NA 1.4 oil immersion objective and the Perfect Focus System for continuous maintenance of focus (Nikon). Green fluorescence images were collected using a 491-nm solid-state laser controlled with an AOTF (Spectral Applied Research) and ET525/50 emission filter (Chroma Technology Corp.). Red fluorescence images were collected using a 561-nm solid-state laser controlled with an AOTF (Spectral Applied Research) and ET620/60 emission filter (Chroma Technology Corp.). All images were acquired with a cooled CCD camera (ORCA AG; Hamamatsu Photonics) controlled with MetaMorph software (version 7.0; Molecular Devices) and archived using ImageJ (National Institutes of Health) and Photoshop CS5 (Adobe). In some cases, linear adjustments were applied to enhance the contrast of images using levels in the image adjustment function of ImageJ and Photoshop.

## Quantification of the number of three-way junctions

Three-way junctions in interphase ER networks were quantified automatically using a custom ImageJ script described previously (*Shemesh et al., 2014*). Three way junctions in mitotic and cytLnp-treated ER networks were counted manually.

## Identification of phosphorylation sites in Lnp

Interphase cytosol containing 5 μM of His$_6$-cytLnp and an energy regenerating system was incubated with buffer or 0.1 mg/mL MBP-cyclin BΔ90 for a 1 hr at room temperature. The reactions were stopped by addition of 1% SDS. After dilution in binding buffer (50 mM Tris pH 7.5, 150 mM NaCl, 20 mM imidazole, 250 mM β-glycerophosphate, 30 mM NEM and 0.63% TX-100), cobalt resin (Clontech) was added to purify His$_6$-cytLnp. After a 30-min incubation at 4°C, the resin was briefly washed with binding buffer three times. His$_6$-cytLnp was eluted in binding buffer with 250 mM imidazole, and subjected to SDS-PAGE. His$_6$-cytLnp bands were excised and phosphorylation sites were identified by mass spectrometry (MS/MS).

For the experiment in *Figure 12C*, similar pull-down experiments were performed as described above, except HA-cytLnp was used and various inhibitors (250 mM β-glycerophosphate, 1% phosphatase inhibitor cocktail 3 (Sigma, P-inhib mix), 1 μM okadaic acid (OA) or 30 mM NEM) were included in the pull-down experiment as indicated in the figure. The input material and proteins eluted from the beads were subjected to immunoblot analysis using *Xenopus* Lnp antibodies.

Human lunapark phosphorylation sites were mapped by stably expressing Lnp-SBP in U2OS cells. For an interphase sample, cells were placed in 0.2% FBS in DMEM overnight. For a mitotic sample, cells were incubated in 10% FBS in DMEM with 100 nM nocodazole overnight; mitotically arrested cells were collected by blowing off the cells from the plate. Cells were lysed in lysis buffer (150 mM NaCl, 20 mM Tris pH 7.5, and protease inhibitors leupeptin, pepstatin and chymostatin) containing 1% TX-100. Streptavidin resin (Thermo Fisher) was used to pull down Lnp. The resin was washed three times with lysis buffer containing 0.1% TX-100, followed by elution with 2 mM biotin. Eluates were TCA precipitated and run on an SDS gel. Silver stained bands were excised and analyzed by MS/MS.

## Immunoprecipitation and pull-down experiments

For the experiment in *Figure 9—figure supplement 1*, 293T cells were infected with lentivirus to express HA- and mCherry-tagged Lnp truncations. The cells were lysed in lysis buffer (150 mM NaCl 20 mM Tris pH 7.5, and protease inhibitors leupeptin, pepstatin and chymostatin) containing 1% TX-100. Cell lysates containing mCherry tagged constructs were normalized to each other using a plate reader that measured fluorescence at a wavelength of 590 nm (Molecular Devices). All incubations

also contained the same total protein concentration, adjusted by adding lysate from uninfected cells, as well as the same amount of HA-tagged bait protein. The samples were incubated with anti-HA affinity matrix (Sigma) for 4 hr at 4°C, and the resin was washed three times in lysis buffer and 0.1% TX-100 before elution with SDS sample buffer. Eluates were analyzed by SDS-PAGE followed by immunoblotting. The same membrane was blotted with anti-mCherry antibodies, stripped of signal using azide solution and reprobed with anti-HA antibody.

For the experiment in *Figure 11—figure supplement 1D*, 10 μL of *Xenopus* membranes and an energy regenerating system were incubated with buffer or 5 μM His$_6$-cytLnp/N1/N2/C-HA for 15 min at room temperature. Binding buffer (20 mM HEPES pH 7.5, 50 mM KCl) was added to a total 200 μL-volume. The membranes were solubilized with 1% Triton X-100 for 30 min and insoluble material was removed by centrifugation at 100,000 g for 30 min at 4°C. Ten μL of the supernatant were saved as input material and the rest was incubated with pre-washed HA agarose resin (Sigma) for 2 hrs at 4°C. After washing of the resin, bound material was eluted with SDS sample buffer by incubation at 37°C for 10 min. The samples were analyzed by SDS-PAGE and immunoblotting with *Xenopus* Lnp antibodies.

For the pull-down experiment shown in *Figure 11—figure supplement 1E*, His$_6$-cytLnp was mixed with buffer or His$_6$-cytLnp-N1/N2/C-HA at a 5:1 ratio for 1 hr at 4°C in binding buffer (20 mM HEPES pH 7.5, 100 mM KCl, 0.05% TX-100). One percent of the reaction was saved as input material and the rest was incubated with pre-washed HA agarose resin for 1 hr at 4°C. After washing of the resin, the bound material was eluted with 200 mM glycine/HCl pH 2.5, 150 mM NaCl, 0.05% TX-100 for 10 min at room temperature. The samples were analyzed by SDS-PAGE and Coomassie blue staining. The pull-down experiments in *Figure 12—figure supplement 1B* were performed in a similar way. The bait was either His6-cytLnp-HA or His$_6$-cytLnp-E-HA and the prey was either Sumo-cytLnp or Sumo-cytLnp-E.

For the experiment in *Figure 12—figure supplement 1A*, interphase cytosol, an energy regenerating system, and His$_6$-cytLnp were incubated with buffer, with buffer and His$_6$-cytLnp-SBP, or with cyclin BΔ90 and His$_6$-cytLnp-SBP for 1 hr at room temperature. The ratio of His$_6$-cytLnp to His$_6$-cytLnp-SBP was 3:1. The volume was adjusted to 200 μL with binding buffer (20 mM HEPES pH 7.5, 100 mM KCl, 0.05% TX-100, 1 μM okadaic acid (Sigma), 1% phosphatase inhibitor cocktail 3 (Sigma), 1 mM DTT). One percent of the input was directly subjected to immunoblotting with the *Xenopus* Lnp antibodies. The rest was incubated with 15 μL of pre-washed magnetic streptavidin resin (Thermo scientific) for 1 hr at 4°C and washed briefly with binding buffer three times. Bound material was eluted with binding buffer containing 2 mM biotin and analyzed by SDS-PAGE and Coomassie blue staining.

## Acknowledgements

We thank Zhonggang Hou for help with CRISPR, Yihong Ye for material, and the Nikon Imaging Center at Harvard Medical School for help with image acquisition and image analysis. We thank Robert Powers and Misha Kozlov for many stimulating discussions, and Junjie Hu, Robert Powers, and Misha Kozlov for critical reading of the manuscript. Songyu Wang is supported by a fellowship from the Charles King Trust. Tom Rapoport is an HHMI Investigator.

## Additional information

### Funding

| Funder | Author |
| --- | --- |
| Howard Hughes Medical Institute | Songyu Wang<br>Hanna Tukachinsky<br>Fabian B Romano<br>Tom A Rapoport |
| Charles A. King Trust | Songyu Wang |

The funders had no role in study design, data collection and interpretation, or the decision to submit the work for publication.

## Author contributions
SW, HT, FBR, TAR, Conception and design, Acquisition of data, Analysis and interpretation of data, Drafting or revising the article

## Author ORCIDs
Tom A Rapoport, http://orcid.org/0000-0001-9911-4216

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
