## [Decision Letter]

Thank you for submitting your article 'Cooperation of the ER-shaping proteins atlastin, lunapark, and reticulons to generate a tubular membrane network' for consideration by *eLife*. Your article has been favorably evaluated by Randy Schekman as the Senior editor and three reviewers, one of whom is a member of our Board of Reviewing Editors and another is Craig Blackstone (Reviewer #2).

The reviewers have discussed the reviews with one another and the Reviewing Editor has drafted this decision to help you prepare a revised submission.

All three reviewers of your paper agree that the article represents an important progress in understanding generation and maintenance of ER morphology by three crucial proteins, Rtn, ATL and Lunapark. It reveals new features of the system such as ER fragmentation upon expression of dominant negative ATLs or overexpression of Rtn4a, tethering role for atlastins in addition to their ability to fuse membranes, the idea that ER structure is in a steady rather than an equilibrium state, which is maintained by a balance between membrane fusion mediated by atlastin and membrane fission.

At the same time, it is the reviewers' opinion that the article will benefit from taking into account the following comments and suggestions.

1) While this paper clearly has substantial data already, additional super-resolution or EM data would make the story even stronger. Particularly, examination of sheet-like structures induced upon lunapark Ab treatment or addition of lunapark fragments in *Xenopus* extracts as in Figure 11 would be useful. Some of what the authors call sheets may appear to be very dense networks of tubules. This issue may be crucial for interpretation of the effect of Lnp.

Accounting for this suggestion is recommended but not required for acceptance of the paper.

2) The most preliminary part of the study is Figure 12 and the reviewers have a number of comments/concerns related to these data. First, the migration shift in panels A and B could be due to modifications other than phosphorylation (e.g., Sumo, Ubiquitin, etc.), and in fact protein bands look quite crisp for what might be a range of phosphorylation states. Second, just because different sites are identified by mass spectrometry doesn't mean all are simultaneously present in the same lunapark protein (raising questions about the relevance of the phosphomimetic construct used). Third, what does the mitotic western look like in the nocodazole treatment arm in panel C? Overall, the reviewers suggest to leave out the entire figure. The authors could consider a much more detailed investigation of the role of mitotic phosphorylation incorporating studies of other proteins involved in ER network formation such as atlastin.

3) The reviewers find the manuscript would benefit from more discussion.

A) Given the suggested tethering function of atlastin, the authors might consider discussion of the published role of SNAREs in ER tubule fusion as has been shown in yeast (JCB 2015 paper by Y. Jun's lab). Maybe alone the atlastins are weak membrane fusogens and predominantly tether?

B) The difference with earlier work should be a little more emphasized. The authors suggest fragmentation was not observed before because fixed cells were used but this is not true in all cases. Also, studies of ER structure in live yeast lacking Sey1 or overexpressing reticulons did not report ER fragmentation. Was fragmentation just missed or what might be the explanation?

C) The manuscript predicts that free ER tubule ends shed vesicles. Is there some way to test this in cells? For example, if ATL is rapidly depleted or inhibited in cells, one would expect to see a significant increase in free ER tubule ends and vesiculation before ER structure is completely disrupted.

D) Finally, the authors may consider relation of the results of the present paper to a model they suggested previously (Shemesh et al. PNAS 2014). According to the 'phase diagram' of Shemesh et al. paper, reduction of the mole fraction of Lnp (denoted there as S/(S+R)), upon a relatively large total amount of the curvature-generating proteins (S+R), drives transition of 3-way junctions to sheets.

Taking this model prediction into account, the scenario of Lnp action might be as follows:

ATL generates 3-way junctions through tubule fusion and remains for a while at place in high local concentration, which may exclude from the junction Rtn and Lnp.

ATL gets inactivated, which is accompanied by its replacement within the junctions by Lnp generating edges with a highly negative edge-line curvature. This would change the micro-structure of the 3-way junction transforming the latter from a shape of constant positive to the shape of a little sheet with flat middle part and the negatively curved edge.

Removal of Lnp by some means, would result in its replacement in the junctions by Rtn, which generates edges with less negative (or vanishing, or positive) edge-line curvature. This would, according to Shemesh's model, drive transformation of 3-way junctions into sheets.

---

## [Author Response]

*All three reviewers of your paper agree that the article represents an important progress in understanding generation and maintenance of ER morphology by three crucial proteins, Rtn, ATL and Lunapark. It reveals new features of the system such as ER fragmentation upon expression of dominant negative ATLs or overexpression of Rtn4a, tethering role for atlastins in addition to their ability to fuse membranes, the idea that ER structure is in a steady rather than an equilibrium state, which is maintained by a balance between membrane fusion mediated by atlastin and membrane fission.*

*At the same time, it is the reviewers' opinion that the article will benefit from taking into account the following comments and suggestions.*

*1) While this paper clearly has substantial data already, additional super-resolution or EM data would make the story even stronger. Particularly, examination of sheet-like structures induced upon lunapark Ab treatment or addition of lunapark fragments in Xenopus extracts as in Figure 11 would be useful. Some of what the authors call sheets may appear to be very dense networks of tubules. This issue may be crucial for interpretation of the effect of Lnp.*

*Accounting for this suggestion is recommended but not required for acceptance of the paper.*

The reviewers raised the question of whether the bright structures in the network generated with the fractionated *Xenopus* system are indeed sheets (Figure 11). These structures were seen with mitotic cytosol or when a dominant-negative cytoplasmic Lnp fragment was added. We assumed that they were sheets because the equivalent conditions with a crude extract resulted in sheets with a normal appearance. However, we admit that the bright areas seen with the fractionated system look different, and we have therefore added a sentence to the subsection 'Lnp is required to form three-way tubular ER junctions in *Xenopus* extracts' to concede that they may not be sheets.

We have tried EM with *Xenopus* extracts, but the images are of poor quality because of non-specific binding of proteins and membranes. The only super-resolution light microscopy techniques that would have enough resolution to distinguish a dense tubular network from sheets would be PALM/STORM. We tried STORM, but encountered problems with the fixation of the *Xenopus* extract and with the sensitivity of the ER network to thiols that are needed to induce fluorophore blinking. We also tried 3D-SIM microscopy, but the current technique does not have enough resolution.

*2) The most preliminary part of the study is Figure 12 and the reviewers have a number of comments/concerns related to these data. First, the migration shift in panels A and B could be due to modifications other than phosphorylation (e.g., Sumo, Ubiquitin, etc.), and in fact protein bands look quite crisp for what might be a range of phosphorylation states. Second, just because different sites are identified by mass spectrometry doesn't mean all are simultaneously present in the same lunapark protein (raising questions about the relevance of the phosphomimetic construct used). Third, what does the mitotic western look like in the nocodazole treatment arm in panel C? Overall, the reviewers suggest to leave out the entire figure. The authors could consider a much more detailed investigation of the role of mitotic phosphorylation incorporating studies of other proteins involved in ER network formation such as atlastin.*

We now provide additional results that support the conclusion that Lnp is mitotically phosphorylated. We therefore prefer to keep Figure 12 in the paper.

We now show that the removal of phosphorylation sites (mutation of Ser and Thr to Ala) prevents the size shift of overexpressed Lnp during mitosis in mammalian cells (Figure 12). Thus, the size shift is likely due to mitotic phosphorylation.

We also provide additional evidence for mitotic phosphorylation in the *Xenopus* system (Figure 12). We show that all modifications of Lnp are lost during a pull-down experiment, but that the less shifted band can be maintained in the presence of phosphatase inhibitors. Both modified Lnp species can be preserved in the presence of N-ethylmaleimide (NEM), so it is possible that the upper band corresponds to an additional modification, as the reviewers suggested. In the mammalian system, Lnp is degraded during mitosis and thus likely ubiquitinated after phosphorylation. We have changed the text in the third paragraph of the subsection “Mitotic phosphorylation of Lnp” to discuss these results.

We now mention that it is unclear whether all phosphorylation sites are equally modified in a given Lnp molecule (subsection “Mitotic phosphorylation of Lnp”).

There is a misunderstanding about the nocodazole experiment shown in Figure 12 (formerly 12C), caused by inaccurate labeling of the figure: nocodazole was used to arrest cells in mitosis (rather than added to interphase cells). We apologize for the mistake.

We also added data (Figure 12) that show that the abundance of overexpressed Lnp in mitotically arrested cells increases upon addition of the proteasome inhibitor bortezomib. No change in abundance is seen with a Lnp mutant in which phosphorylation sites are mutated to alanine. These data support the idea that Lnp is first phosphorylated during mitosis and then degraded by the proteasome, at least in mitotically arrested cells.

*3) The reviewers find the manuscript would benefit from more discussion.*

*A) Given the suggested tethering function of atlastin, the authors might consider discussion of the published role of SNAREs in ER tubule fusion as has been shown in yeast (JCB 2015 paper by Y. Jun's lab). Maybe alone the atlastins are weak membrane fusogens and predominantly tether?*

We now mention this paper in the Discussion (fourth paragraph). However, we do not agree with the conclusions drawn by Y. Jun’s lab, who claim that Sey1p and ER SNAREs function in the same pathway. The best argument against their model is that mutations in SEY1 and ER SNAREs have synthetic effects, indicating that they function in parallel pathways. There is no question that Sey1 and Atl can mediate fusion on their own.

*B) The difference with earlier work should be a little more emphasized. The authors suggest fragmentation was not observed before because fixed cells were used but this is not true in all cases. Also, studies of ER structure in live yeast lacking Sey1 or overexpressing reticulons did not report ER fragmentation. Was fragmentation just missed or what might be the explanation?*

We do believe that fixation is the main reason for why ER fragmentation was missed previously. “ER discontinuity” (ER fragmentation) has been observed by Orso et al. in *Drosophila* treated with ATL RNAi constructs, and by Moss et al. (2011), who saw “punctate ER” in fixed cells when cytATL was overexpressed. We are not aware of earlier papers in which ATL function was blocked and live-cell imaging was performed.

The situation in *S. cerevisiae* may be different from that in mammalian cells. Fragmentation of the ER may not occur because ER SNAREs provide an alternative pathway of ER fusion in this organism. It is also possible that some ER fragmentation goes unnoticed because of the small cell size and difficulties in imaging.

We have changed the wording of the last paragraph of the subsection “ATL is required to maintain tubules and junctions in mammalian cells” and now cite the Moss et al. paper (the Orso et al. paper was cited before).

*C) The manuscript predicts that free ER tubule ends shed vesicles. Is there some way to test this in cells? For example, if ATL is rapidly depleted or inhibited in cells, one would expect to see a significant increase in free ER tubule ends and vesiculation before ER structure is completely disrupted.*

It is impossible to test the shedding of vesicles in cells, if only because there is no way to rapidly deplete or inhibit ATL. In *Xenopus* extracts, we observed that the inhibition of ATL by cytATL or GTPγS leads to short wiggling tubules and vesicles, but even in this system, it is difficult to demonstrate that the tubules disassemble by shedding vesicles. Our hypothesis is thus mostly based on the fact that free tubule ends are rarely seen in mammalian cells or *Xenopus* extracts.

*D) Finally, the authors may consider relation of the results of the present paper to a model they suggested previously (Shemesh et al. PNAS 2014). According to the "phase diagram" of Shemesh et al. paper, reduction of the mole fraction of Lnp (denoted there as S/(S+R)), upon a relatively large total amount of the curvature-generating proteins (S+R), drives transition of 3-way junctions to sheets.*

*Taking this model prediction into account, the scenario of Lnp action might be as follows:*

*ATL generates 3-way junctions through tubule fusion and remains for a while at place in high local concentration, which may exclude from the junction Rtn and Lnp.*

*ATL gets inactivated, which is accompanied by its replacement within the junctions by Lnp generating edges with a highly negative edge-line curvature. This would change the micro-structure of the 3-way junction transforming the latter from a shape of constant positive to the shape of a little sheet with flat middle part and the negatively curved edge.*

*Removal of Lnp by some means, would result in its replacement in the junctions by Rtn, which generates edges with less negative (or vanishing, or positive) edge-line curvature. This would, according to Shemesh's model, drive transformation of 3-way junctions into sheets.*

We thank the reviewers for suggesting this interesting model. However, the model would require that Lnp be relatively abundant (which it is not) and that we ignore the overexpression results, which show that Lnp forms sheet-like structures. We therefore feel that our results are not really consistent with the proposed idea. At this point, all models of Lnp function remain rather speculative.